# LFQA-E: Carefully Benchmarking Long-form QA Evaluation

**Yuchen Fan**[1,2,*]**, Chen Ling**[5,*]**, Xin Zhong**[6,*]**, Shuo Zhang**[4]**, Heng Zhou**[2]**, Yuchen Zhang**[2]**,
Mingyu Liang**[4]**, Chengxing Xie**[2]**, Ermo Hua**[3]**, Gang Chen**[3]**, Zhizhou He**[3]**, Cheng Huang**[3]**,
Ning Ding**[3,†]**, Bowen Zhou**[2,3,†]

[1] Shanghai Jiao Tong University    [2] Shanghai AI Lab    [3] Tsinghua University
[4] Beijing University of Posts and Telecommunications    [5] Zhejiang University
[6] University of Tokyo
[*]Equal contributions    [†]Corresponding author

`yuchenfan48@gmail.com`  `zhoubowen@tsinghua.edu.cn`

## Abstract

Long-Form Question Answering (LFQA) involves generating comprehensive, paragraph-level responses to open-ended questions, which poses a significant challenge for evaluation due to the richness of information and flexible response format. Existing LFQA-evaluation benchmarks often lack reference answers and are limited in size and topic coverage, reducing their reliability. To address this gap, we introduce LFQA-E, a well-constructed, multilingual, and reference-based benchmark designed to rigorously evaluate automatic metrics for LFQA. LFQA-E comprises 1618 questions and 7323 pairwise comparisons across 15 topics, drawn from diverse sources such as online queries and examination questions, thereby enabling a comprehensive assessment of evaluation metrics. We examine five categories of metrics, encompassing 17 specific methods, using LFQA-E. The results demonstrate that none of the existing automatic metrics perform comparably to human judgments, highlighting their inability to capture the dense information in long-form responses. Furthermore, we present a detailed analysis of the failure cases and the generalization capacity of these metrics, offering insights to guide the future development of LFQA evaluation methods. The benchmark and code are available at `https://github.com/YuchenFan48/LFQA-E`.

## 1 Introduction

Long-form Question Answering (LFQA) (Fan et al., 2019) targets at generating in-depth, paragraph-level responses to open-ended questions. It requires models to have comprehensive domain-specific knowledge or use evidence from retrieved documents (Nakano et al., 2022; Akash et al., 2023) to provide relevant and accuracy answers. Despite efforts to enhance the quality of long-form answers, developing automatic and reliable evaluation metrics for LFQA is still underexplored.

Evaluating long-form answers presents significant challenges, as evaluators must possess comprehensive domain knowledge. Previous manual evaluations typically employed crowd-sourced workers for annotation. However, their limited domain expertise inevitably compromises reliability. In contrast, expert annotation would ensure higher quality, while the cost of employing experts to annotate large-scale datasets is prohibitive. Consequently, automatic evaluation metrics are essential. In automatic evaluation of LFQA, ROUGE (Lin, 2004) has been widely adopted. However, Krishna et al. (2021) argue that ROUGE provides limited informativeness in long-form contexts, weakening its reliability. With the advancement of LLMs (OpenAI, 2023; 2024) and Large Reasoning Models (LRMs) (DeepSeek-AI et al., 2025a; Zhang et al., 2025), numerous studies have leveraged these models to develop evaluation metrics through various approaches (Chang et al., 2023), including prompting (Wei et al., 2023; Zhou et al., 2025), fine-tuning (Li et al., 2023; Fan et al., 2024a; Jiang et al., 2024), and training LLMs as Reward Models (RMs) (Liu et al., 2024a; Chen et al., 2025),

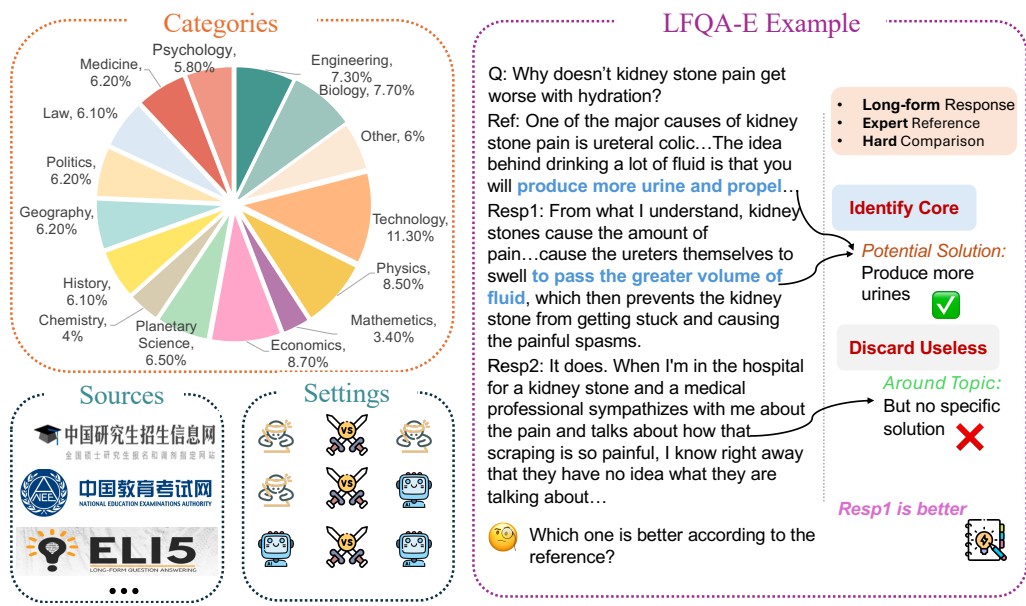

Figure 1: The figure shows the overview of LFQA-E. The left side displays the categories, sources, and three settings, showcasing its diversity. The right side illustrates an example of LFQA-E.

either generative or scalar-based. Despite the advances of evaluation metrics, determining which metrics are most effective and best aligned with human judgment for LFQA evaluation requires systematic verification and benchmarking.

Previous benchmark for LFQA evaluation (Xu et al., 2023) samples records from reddit/ELI5, hiring experts to annotate the better one between two responses without references, and test alignment between automatic evaluation metrics and expert labels. However, their benchmark has several limitations: **1) Lack of authorized references** A reference answer provides a baseline for assessing whether a response covers key details and maintains factual accuracy. Without ground-truth references, the comparison between metrics may be unfair, and evaluations without clear criteria or rubrics are inherently unreliable. **2) Limited diversity** The benchmark consists of only *260* examples, all in English, constraining its linguistic and topical diversity. Moreover, it treats the comparison as an A/B task, but in real scenarios, a "tie" option always exists.

To fill the gap, we introduce LFQA-E, towards evaluating the ability of different metrics. 1) To evaluate whether current automatic evaluation metrics can select a better one from two nuanced responses, we gather references that are examined by the experts, and judge based on them. To ensure the difficulty, we choose human responses based on their upvotes or their scores, and model responses based on two models with comparable capabilities. 2) To analyze the systematic differences in validity among evaluation metrics, we rigorously assess the performance from several aspects. First we evaluate them based on three settings, i.e, human vs human ($h$ v. $h$), human vs model ($h$ v. $m$), and model vs model ($m$ v. $m$). Moreover, we collect multilingual responses, i.e, English and Chinese, and multiple domain-specific responses, e.g., Engineering, Law, Medicine, to ensure a thorough analysis. 3) To prevent data contamination, we collect data from offline examination, i.e., College Entrance Examination Simulation Questions (CEESQ) and Postgraduate Entrance Examination Questions (PEEQ) and online platform questions (reddit/ELI5) from the recent half-year. The overview of LFQA-E Benchmark is shown in Figure 1.

Using LFQA-E, we critically assess the efficacy of 17 evaluation metrics. The experimental results show that current leading evaluation metrics fail to capture core information as human beings from verbose responses when differentiating the better one between two responses with similar quality. Furthermore, we provide analysis on why automatic evaluation metrics fail in LFQA evaluation and find the misalignment between evaluation metrics. Lastly, we try TTRL (Zuo et al., 2025) to improve the evaluation performance of model-based metrics and provide some actionable insights.

## 2 RELATED WORK

**Development of LFQA**  LFQA (Fan et al., 2019) requires models to generate paragraph-level responses to open-ended questions which is more complex compared to datasets like SQuAD (Rajpurkar et al., 2016), TriviaQA (Joshi et al., 2017), and NarrativeQA (Kočiský et al., 2017), where answers are primarily words or phrases extracted directly from documents. In LFQA, models must generate comprehensive yet correct responses based on their knowledge or existing evidence documents. Several studies have analyzed the discourse structure of long-form answers (Xu et al., 2022) and have sought to enhance the performance on LFQA (Chen et al., 2023; Akash et al., 2023).

**Evaluation of LFQA**  The automatic evaluation of LFQA remains challenging and underexplored. For human annotation, HURDLES (Krishna et al., 2021) and WEBGPT (Nakano et al., 2022) employ A / B testing, where crowd-sourced annotators are instructed to choose the best of two candidate answers. Since annotation of LFQA requires high expertise, the results of crowd-sourced workers may be unreliable. To address the gap, Xu et al. (2023) employs experts for annotation, and tests several evaluation metrics, such as ROUGE (Lin, 2004), BERTScore (Zhang et al., 2020), and BARTScore (Yuan et al., 2021), on an expert-annotated dataset. Their findings validated that no existing metrics fully align with human judgment. However, the dataset they used lacks expert-written references, which are sourced from Reddit/ELI5, and is limited in scale, comprising only about 260 samples. More recently, since the development of LLMs and LRMs, many work uses them for evaluation of free-form answers, using prompt (Wei et al., 2023), fine-tuning using specific data (Liu et al., 2023), and reinforcement-learning (Chen et al., 2025).

## 3 METHODOLOGY

### 3.1 OVERVIEW

To reasonably test the evaluation ability of different metrics for LFQA when having a reference, we introduce LFQA-E, a multilingual and comprehensive benchmark composed of different topics and questions. LFQA-E consists of the Chinese version LFQA-E-ZH and the English version LFQA-E-EN. Table 1 shows its overview. It includes 1618 questions and 7323 comparisons, consisting of 1193 comparisons in Chinese and 6130 comparisons in English. It spans 15 topics, ranging from history to engineering, ensuring its diversity. LFQA-E comprises expert-annotated references for fair comparison and nuanced responses. Therefore, it is naturally a hard yet reasonable benchmark for LFQA evaluation.

**Reference-Based Evaluation**  For LFQA-E, references are sourced from academic examinations or widely discussed questions in Reddit/ELI5. After being reviewed by experts with relevant academic backgrounds, these references are ensured to cover all the key points needed to answer the question. This provides a baseline for evaluation metrics to look up and provide a more precise comparison.

Table 1: Detailed statistics of LFQA-E. **Avg Que. Lens**, **Avg Ref. Lens**, **Avg Res. Lens** corresponds to question lengths, reference lengths, and response lengths, respectively.

|  | LFQA-E-EN | LFQA-E-ZH |
|---|---|---|
| **# Topics** | 9 | 6 |
| **# Questions** | 1026 | 592 |
| **# Comparisons** | 6130 | 1193 |
| **# Avg Que. Lens** | 13.4 | 24.6 |
| **# Avg Ref. Lens** | 299.1 | 187.2 |
| **# Avg Res. Lens** | 245.0 | 308.3 |
| **Annotate** | Expert | Expert |

**Difficult Comparisons**  All the questions contained in LFQA-E have been carefully examined by domain experts to ensure it is answerable and clear to understand. We ensure that models have not seen the data by collecting data from recent examinations and forum questions. The responses are collected from human-written responses, with close scores or upvotes, and model responses generated by comparable LLMs. Therefore, it is hard to distinguish the better one at a glance.

**Diverse Benchmark**  We collect 1618 questions and 7323 comparisons in 15 distinct domains, from natural science to social science, to guarantee a diverse and representative benchmark. Also, LFQA-E is multilingual, consisting of examples in both Chinese and English. Moreover, LFQA-E includes three kinds of comparisons, guaranteeing the comprehensibility of the benchmark.

## 3.2 DATA PROCESSING

**Data Collection**    For LFQA-E-ZH, we source our data from CEESQ and PEEQ, where questions, references, and scoring schemas are developed by domain experts, including teachers and professors from high schools and colleges. The records are sampled from 2024 and are based on local examinations restored from PDF files that have not been submitted to online platforms. These questions cover diverse subjects, including politics, history, medicine, psychology, law, and geography. We provide two examples of CEESQ and PEEQ in Appendix D.1. For LFQA-E-EN, data is sourced from Reddit/ELI5, where each question is explained without specialized terminology or complex concepts, and we use the top-ranked answer as our candidate reference. For LFQA-E-ZH, to prevent overlap with potential training data, we avoid using data from actual College Entrance Examinations. All problems are sourced from PDF files and have not been uploaded online. Additionally, the questions captured from ELI5 are all from the past six months. We provide a benchmark contamination study in Sec 5.5. To ensure that all questions are clear and answerable, we instruct GPT-4o with temperature=0.7 to filter out questions with unclear descriptions. The instruction used is listed in Appendix E.1. We conducted an experiment on the effectiveness of leveraging GPT-4o as a filter, and the results reveal that GPT-4o demonstrates superior performance for this task, achieving 97% accuracy. Subsequently, to ensure our references contain all information needed to answer the questions, we submit the remaining data for expert annotation. The annotation guidelines are presented in Table 29. Each reference was annotated by two annotators, and references were discarded if either annotator labeled them as invalid. The Cohen's kappa coefficient is 0.78, indicating substantial inter-annotator agreement. Through this process, we obtained 1618 questions.

**Human Response Collection**    For LFQA-E-ZH, we gather examination papers primarily in image format and employ Optical Character Recognition (OCR) systems to extract student responses. Specifically, we choose student answers with close scores to ensure the comparison difficulty. The OCR is conducted using the Volcano Engine API. For LFQA-E-EN, we collect responses from the forum section of the corresponding question. Also, we select answers within the many-voted yet close up-votes to make them hard to differentiate. However, the responses we collect for LFQA-E-ZH are mainly written during examination, it is concise and well structured, and the responses we collect for LFQA-E-EN include some special characters like URLs, which deteriorate our data quality. To handle it, we use GPT-4o to paraphrase and clean our human responses. The instruction we used is shown in Appendix E.1. We randomly sample 100 records and annotate the paraphrased responses to validate the performance of GPT-4o, with the results in Appendix B.1, which indicates that leveraging LLMs doesn't introduce any error. We also provide a case study in Appendix D.2.

**Model Response Generation**    When generating model responses, we focus on evaluating whether LLMs can understand the semantic meaning of texts well and properly select the better response. Therefore, we do not impose extremely strict requirements on answer quality. Instead, we ensure the difficulty of LFQA-E BENCH by selecting models with similar ranking in the LMSYS Arena (Chiang et al., 2024; Zheng et al., 2023; 2024). On account of responses generated by stronger models like GPT-4o or Claude-4-sonnet will pose a great challenge for our annotators to differentiate, increasing the cost of annotation, we leverage Llama-3-8B-Instruct (Dubey et al., 2024) and GPT-3.5-turbo (OpenAI, 2023) for response generation. For model-generated answers, we use *"Generate reasonable answers to the following questions. Use references or examples if needed"* to prompt LLMs. The generation temperature is set to 1.0 to encourage diverse and creative responses.

## 3.3 HUMAN ANNOTATION

**Annotator Decision**    We hire 10 annotators from relevant aspects or who have taken relevant courses. Then we provide them with clear annotation recipes for better quality control. The annotation recipe is in Appendix F. Each annotator receives 2\$ for annotating a question, including 4-6 comparisons. To ensure the effectiveness of the annotation, we pre-annotate a subset consisting of 35 records. The annotator will start their work until they reach a 90% consistency on the subset.

**Annotation Setting**    Guided by Xu et al. (2023), our evaluation criteria mainly focus on factuality and completeness according to the reference, since almost all responses we collect are already very fluent. Unlike typical A/B testing, our method employs a triple-choice format, giving a tie option,

to better capture the subtle differences between answers, as they often show comparable levels of information overlap with the reference while the other information is useless or verbose according to the central topic which can de dropped without hindering the comprehension of the response.

Table 2: Performance of evaluation metrics on LFQA-E. The largest value is denoted in **bold**. We provide the results of other models in Appendix B.4.

| Model | LFQA-E-EN | | LFQA-E-ZH | | $\text{Avg}_{F1}$ | $\text{Avg}_{Acc}$ |
|---|---|---|---|---|---|---|
| | F1 | Accuracy | F1 | Accuracy | | |
| Static Evaluation Metric | | | | | | |
| Human Baseline | 77.7 | 83.3 | 68.9 | 76.5 | 73.3 | 79.9 |
| Length | 26.0 | 42.8 | 33.5 | 52.6 | 30.8 | 47.7 |
| ROUGE | 37.5 | 55.5 | 34.0 | 49.7 | 35.8 | 52.6 |
| BERTScore | 35.9 | 54.1 | 36.6 | 52.4 | 36.3 | 53.3 |
| LLMs-based Evaluation Metric | | | | | | |
| Qwen2.5-32B-Instruct | 45.8 | 63.5 | 41.8 | 56.7 | 43.8 | 60.1 |
| Qwen2.5-72B-Instruct | 43.1 | 61.2 | 39.0 | 53.0 | 41.1 | 57.1 |
| Llama3.1-70B-Instruct | 42.5 | 59.6 | 29.4 | 30.7 | 36.0 | 45.2 |
| GPT-4o | **46.4** | 61.7 | 42.6 | 53.2 | **44.5** | 57.5 |
| DeepSeek-V3 | 39.3 | 57.9 | 41.1 | 53.8 | 40.2 | 55.9 |
| RM-based Evaluation Metric | | | | | | |
| Skywork-Reward-Llama | 37.3 | 54.4 | 38.2 | 53.6 | 37.8 | 54.0 |
| Skywork-Reward-Gemma | 37.5 | 56.0 | 33.0 | 48.3 | 35.3 | 52.2 |
| RM-R1-Qwen2.5-Instruct-14B | 36.4 | 64.9 | 35.1 | 51.9 | 35.8 | 58.4 |
| RM-R1-DeepSeek-Distilled-Qwen-14B | 43.9 | 65.7 | 36.7 | 53.2 | 40.3 | 59.5 |
| LRM-based Evaluation Metric | | | | | | |
| o1-mini | 45.9 | 62.9 | **45.2** | **58.9** | 45.6 | **60.9** |
| Deepseek-R1 | 42.9 | 59.6 | 42.4 | 57.8 | 42.7 | 58.7 |
| Trained Evaluation Metric | | | | | | |
| Auto-J-6B-bilingual | 46.0 | **66.8** | 35.4 | 51.9 | 40.7 | 59.4 |
| Prometheus-7B-v2.0 | 41.8 | 64.2 | 34.1 | 50.1 | 38.0 | 57.2 |
| M-Prometheus-14B | 41.6 | 60.8 | 33.9 | 49.4 | 37.8 | 55.1 |

**Annotation Process** The annotators assess two responses against a given reference and select the more informative and complete answer or declare a "tie" if both are of similar quality. During the process, we treat a piece of information as the basic unit as FActScore (Min et al., 2023). Initially, annotators extract the key information needed to answer the question from the provided reference and check whether the responses under evaluation contain similar statements. Then, they will select a better one based on the overlapped information. To minimize bias and subjectivity, each record is annotated by two independent reviewers. Each comparison takes around 7 minutes to annotate. For some hard-to-differentiate comparisons, detailed justification is saved to help understand. After annotation, we find the Cohen's kappa correlation of inter-annotator agreement is approximately 0.65, indicating a substantial agreement. We show a screenshot in Figure 5.

## 4 EXPERIMENTS

### 4.1 MODELS

We evaluate various metrics on LFQA-E-EN and LFQA-E-ZH respectively, including: **Static Metrics:** We use Length-orientation, ROUGE-1 (Lin, 2004) and BERTScore (F1) Zhang et al. (2020) since they are widely used as the evaluation metric for LFQA. **LLMs:** We select Qwen2.5-32B-Instruct (Qwen et al., 2025), Qwen2.5-72B-Instruct, Llama-3.1-70B-Instruct (Dubey et al., 2024), Deepseek-V3 (DeepSeek-AI et al., 2025b), and GPT-4o (OpenAI et al., 2024). **LRMs:** Considering the high time complexity and cost, we use o1-mini and Deepseek-R1 (DeepSeek-AI et al., 2025a). **RMs:** We test on Skywork-Reward-Gemma-2-27B-v0.2 (Liu et al., 2024a), Skywork-Reward-Llama-3.1-8B-v0.2 considering their leading position on Reward Bench (Lambert et al., 2024). We also test RM-R1-Qwen2.5-Instruct-14B (Liu et al., 2024b) and RM-R1-Deepseek-Distilled-Qwen-14B since they represent another paradigm of reward models. We refer to Skywork-Reward-

Gemma-2-27B-v0.2 and Skywork-Reward-Llama-3.1-8B-v0.2 as Skywork-Reward-Gemma and Skywork-Reward-Llama for simplification. **Evaluation-Specific Models:** There are some models trained to be evaluation models. Among these models, we select Auto-J-Bilingual (Li et al., 2023), Prometheus-7B-v2.0, and M-Prometheus-14B (Kim et al., 2024).

## 4.2 IMPLEMENTATION DETAILS

We evaluate all the metrics in both LFQA-E-EN and LFQA-E-ZH. We use Jieba cut for ROUGE-zh. For BertScore, we use roberta-large for LFQA-E-EN evaluation and bert-base-chinese for LFQA-E-ZH. We set the temperature at 1.0 for all LLM-based evaluation metrics to encourage diverse responses. We show the results of temperature $= 0$ in Sec 5.1. The prompts we used are shown in Appendix E.2. We further conduct a prompt-sensitivity analysis in Appendix B.2. For models with specific training templates, we adopt them. We include references for models to look up in all our settings. We use accuracy and macro-F1 as our indicators:

$$\text{Acc} = \frac{1}{N} \sum_{i=1}^{N} \mathbb{I}(\text{pred}_i = \text{label}_i) \tag{1}$$

$$\text{F1}_{\text{macro}} = \frac{1}{||\mathcal{C}||} \sum_{c \in \mathcal{C}} \left( 2 \cdot \frac{P_c R_c}{P_c + R_c} \right) \tag{2}$$

where $\mathcal{C} = \{A, B, \text{tie}\}$. For LLM-based methods, we include a "tie" option in the instruction, while for other methods that return a scalar, we round the scalar to 3 decimal places. For the human baseline, we hire another 3 annotators with doctor's degrees to ensure the quality. The final scores are obtained by averaging annotators' results. We use the annotation recipe as previously claimed.

## 4.3 MAIN RESULTS

Table 2 lists our results. The overall low accuracies and F1-scores of all evaluation metrics indicate the challenge LFQA-E poses to current models and methods. We also provide a cost analysis in Appendix B.6.

**Comparison Between Metrics** Though none of the evaluation metrics achieves a high performance on LFQA-E, we observe that scaling model size doesn't definitely yield a better result. For example, Qwen2.5-32B-Instruct beats Qwen2.5-72B-Instruct by $3\%$. What's more, LRMs show a great performance compared with LLMs, thanks to their long CoT and extended thinking. RM-based evaluation metrics don't show promising results when generalizing to LFQA evaluation, perhaps because they are trained to give a better one between two responses, renouncing the "tie" option. We will analyze further and give a fairer comparison in Section B.5.

Table 3: Performance of different models on comparisons that humans labeled as *tie*. The largest value in each column is in **bold**.

| Model | LFQA-E-EN | LFQA-E-ZH |
|---|---|---|
| Deepseek-V3 | 1.8 | 10.2 |
| Qwen2.5-32B-Instruct | 7.2 | 7.5 |
| Qwen2.5-72B-Instruct | 2.6 | 3.6 |
| Llama-3.1-70B-Instruct | 5.0 | 7.7 |
| o1-mini | 7.1 | 14.1 |
| Deepseek-R1 | 7.4 | 7.2 |
| GPT-4o | **9.2** | **14.6** |

**Comparison Between Indicators** All evaluation metrics struggle to give a tie as good as human beings. Table 3 indicates that among the evaluation metrics we test, the best result is just $9.2\%$ for LFQA-E-EN and $14.6\%$ for LFQA-E-ZH. Observing the responses, we find that they are too conservative to claim two responses are of equal quality. This explains why accuracy is always larger than Macro-F1. The low accuracy on tie comparison reflects the difficulty of LFQA-E again.

**Specialized Tuned models Help Boost Performance.** Observing the table above, we find that tuning a base model to be a robust generative reward model helps in LFQA evaluation. Moreover, when changing the base model to a strong reasoning model, the performance gains continually. In addition, SFT models can also achieve performance comparable with models of larger sizes. These phenomenon indicates that tuning is essential for LFQA evaluation.

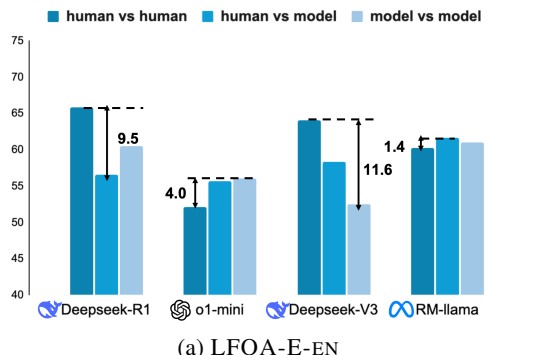 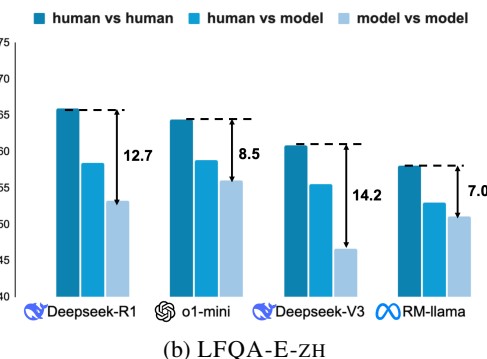

(a) LFQA-E-ᴇɴ                                   (b) LFQA-E-ᴢʜ

Figure 2: Performance of different models on our three settings on LFQA-E.

## 5 ANALYSIS

### 5.1 TEMPERATURE MATTERS

To ablate the effect of temperature on our validation, we further experiment using a temperature equals 0.0 for deterministic results. The result in Table 4 shows a noticeable shift in metrics. For instance, the performance of LLM-based Evaluation Metrics such as Qwen2.5-32B-Instruct and GPT-4o are higher compared to when the temperature is set to 1.0. This suggests that deterministic behavior likely reduces the model's variability in responses, leading to more consistent evaluation results. However, the LRM-based evaluation metrics show a significant drop, indicating that models relying on more exploration perform worse or even collapse under deterministic conditions.

Table 4: Performance of metrics on LFQA-E when temperature is set to 0. The largest value is denoted in **bold**.

| Model | LFQA-E-ᴇɴ | | LFQA-E-ᴢʜ | | $\mathbf{Avg}_{F1}$ | $\mathbf{Avg}_{Acc}$ |
|---|---|---|---|---|---|---|
| | F1 | Accuracy | F1 | Accuracy | | |
| LLMs-based Evaluation Metric | | | | | | |
| Qwen2.5-32B-Instruct | 46.0 | **64.5** | 40.0 | 54.0 | 43.0 | **59.3** |
| Qwen2.5-72B-Instruct | 41.8 | 58.6 | 38.5 | 52.7 | 40.2 | 55.7 |
| Llama3.1-70B-Instruct | 36.8 | 53.5 | 40.5 | 50.4 | 38.6 | 52.0 |
| GPT-4o | **49.5** | 63.1 | **43.9** | 52.7 | **46.7** | 57.9 |
| DeepSeek-V3 | 40.1 | 57.8 | 40.9 | **55.1** | 40.5 | 56.5 |
| LRM-based Evaluation Metric | | | | | | |
| o1-mini | 43.4 | 56.2 | 4.3 | 5.8 | 23.9 | 31.0 |
| Deepseek-R1 | 2.0 | 2.7 | 32.1 | 45.3 | 17.1 | 24.0 |

### 5.2 EVALUATION METRICS CAN'T EXCEL AT ALL SETTINGS.

To have a thorough understanding of whether the model evaluates human response or model response differently, we experiment on a different group of LFQA-E. We break it into three groups, i.e., *h v. h*, *h v. m*, and *m v. m*, where *h* indicates human response and *m* represents model response, and see the accuracy changes. The results are listed in Figure 2a for LFQA-E-ᴇɴ and Figure 2b for LFQA-E-ᴢʜ. We can observe that for many evaluation metrics, there exists a huge difference between different comparison settings. In LFQA-E-ᴇɴ, the RMs show steady ability while others exhibit degradation when model responses are introduced. In LFQA-E-ᴢʜ, all the metrics show a drastic accuracy decline under *m v. m*, with a maximum drop of $14.2\%$ from Deepseek-V3. This further validates our assumption that current evaluation metrics can't differentiate between two nuanced responses. we also show the performance across subjects in Appendix B.3.

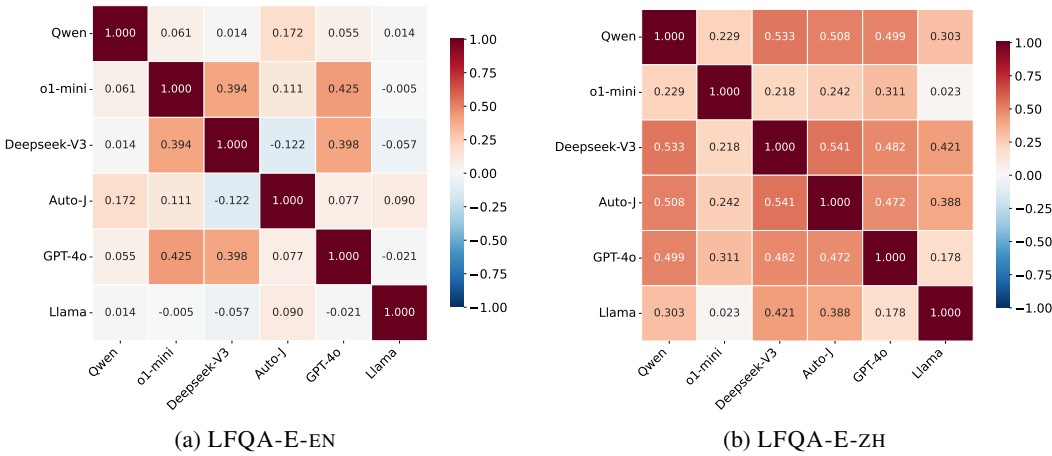

(a) LFQA-E-EN  (b) LFQA-E-ZH

Figure 4: The Cohen's Kappa Correlation Matrix in LFQA-E.

## 5.3 REASONS EVALUATION METRICS FAIL WHEN EVALUATING LONG-FORM RESPONSES.

**For LM-based Evaluation Metric**  We observe the outputs of several LLMs and find that almost all errors arise from the following aspects.

- *Keypoints Identification Error*: The model fails to correctly identify and separate bullet keypoints or enumerated lists in responses, leading to poorly structured answers.
- *Irrelevant/Incorrect Information Error*: The model does not penalize or filter out irrelevant or factually incorrect details in its responses, reducing accuracy.
- *Contradiction Error*: During reasoning, the model generates inconsistent or contradictory statements due to factual hallucinations.
- *Formatting Error*: The model produces responses with an improper format.

We show the probability of each error occurring in Figure3. We choose Deepseek-V3 and o1-mini for representation. *Point Identification Error* and *Irrelevant/Incorrect Information Error* happen most time, indicating the relatively low inherent ability for LMs when evaluating long-form answers.

**Static Evaluation Metrics**  These methods simply leverage word-level or embedding-level similarities, which scratch on surface when evaluating. As described in Fan et al. (2024b), when considering evaluating two long responses around a topic, there may be many words overlapping. Also, overly long responses dilute semantics, making originally important key information trivial, so metrics fail to consider informativeness, but only focus on similarity.

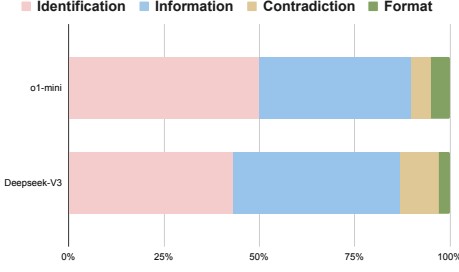

Figure 3: Percentage of error types for LMs on the LFQA-E dataset.

## 5.4 DIFFERENT EVALUATION METRICS DON'T AGREE WITH EACH OTHER.

To find whether there is a correlation between different evaluation metrics, we observe detailed evaluation results. Specifically, we select ROUGE, Qwen2.5-32B-Instruct (simplified as Qwen), GPT-4o, Skywork-Reward-Llama (simplified as Llama), o1-mini, and Auto-J-6B-bilingual (simplified as Auto-J), considering their relatively better performance on LFQA-E. Figure 4a and 4b show the results. We observe that neither of the two metrics achieves a high correlation, indicating two metrics may contradict each other to a large degree. There are even some negative correlations between the two metrics under LFQA-E-EN. This phenomenon further illustrates that there is no stable evaluation result across different metrics.

## 5.5 BENCHMARK CONTAMINATION ANALYSIS

To examine whether our evaluation benchmark has potential contamination from pretraining corpora, we conducted perplexity (PPL) and n-gram overlap, the same as Xu et al. (2024), using two widely adopted open-source instruction-tuned models: `Qwen2.5-7B-Instruct` and `Llama-3.1-8B-Instruct`. The underlying intuition is that if evaluation data is memorized during pretraining, the models would exhibit abnormally low perplexity and high n-gram overlap on the evaluation benchmark.

**Perplexity Analysis.** Table 5 reports the PPL values on both English and Chinese subsets. For `Qwen2.5-7B-Instruct`, the PPL is 11.60 (en) and 11.73 (zh). For `Llama-3.1-8B-Instruct`, the PPL is 11.70 (en) and lower at 7.21 (zh), which all fall within tolerable limits and low data contamination potential.

Table 5: Perplexity (PPL) on the benchmark.

| Model | en | zh |
|---|---|---|
| Qwen2.5-7B-Instruct | 11.60 | 11.73 |
| Llama-3.1-8B-Instruct | 11.70 | 7.21 |

Table 6: $n$-gram accuracy on the benchmark.

| Model | en | zh |
|---|---|---|
| Qwen2.5-7B-Instruct | 0.030 | 0.047 |
| Llama-3.1-8B-Instruct | 0.025 | 0.093 |

$n$**-gram Overlap.** We further computed n-gram exact match accuracy between the benchmark and model generations. As shown in Table 6, the overlap scores remain low across both models, e.g., 0.025–0.030 in English and 0.047–0.093 in Chinese. These values are significantly below contamination thresholds observed in prior work, reinforcing the view that large-scale memorization is unlikely to happen in our benchmark.

Overall, both analyses suggest that the benchmark is not heavily contaminated. The English subset appears relatively safe across both models. For the Chinese subset, while Llama-3.1-8B-Instruct shows somewhat lower perplexity and higher n-gram overlap, though n-gram higher than expected, remain within acceptable bounds and don't indicate severe memorization.

## 5.6 COMPARISON WITH ESTABLISHED BENCHMARKS

To demonstrate the value and distinctiveness of our LFQA-E benchmark, we conduct a comparative analysis against several established baselines using frontier models. Our findings consistently show that LFQA-E presents a significantly greater challenge, thereby better differentiating the capabilities of state-of-the-art models.

Table 7: Performance of TTRL.

| Model | LFQA-E-EN |
|---|---|
| **Qwen2.5-3B-Instruct** | |
| CoT | 49.6 |
| Structured Prompt | 59.7 |
| TTRL | 63.9 |
| TTRL + Clip Higher | 66.5 |
| **Qwen2.5-7B-Instruct** | |
| CoT | 53.3 |
| Structured Prompt | 60.6 |
| TTRL | 68.2 |
| TTRL + Clip Higher | 68.6 |

**General LLM Evaluation.** First, we compare LFQA-E with the expert-annotated benchmark from Xu et al. (2023), denoted as "Expert" and Feedback-Bench Kim et al. (2024). As shown in Table 8, all leading models achieved substantially lower scores on LFQA-E compared to the baselines. This performance gap validates that LFQA-E probes deeper into the models' reasoning and evaluation capabilities rather than merely surfacing superficial knowledge.

**Reward Model Evaluation.** We further extend our comparison to specialized reward model benchmarks, including RM-Bench Liu et al. (2024c) and Reward-Bench Lambert et al. (2024). The results, presented in Table 9, reaffirm our conclusion. The consistent performance drop on LFQA-E across various reward models highlights its robustness and difficulty, even for systems specifically designed for evaluation tasks.

Table 8: Performance comparison between LFQA-E and general evaluation benchmarks.

| Model | Feedback-Bench | Expert | LFQA-E |
|---|---|---|---|
| Qwen2.5-32B-Instruct | 86.8% | 74.2% | 60.1% |
| Qwen2.5-72B-Instruct | 94.4% | 72.7% | 57.1% |
| GPT-4o | 89.2% | 70.0% | 57.5% |
| GPT-5 | 75.6% | 68.9% | 62.5% |
| DeepSeek-V3 | 83.6% | 65.8% | 55.9% |
| DeepSeek-R1 | 83.6% | 68.9% | 58.7% |

Table 9: Comparison with specialized reward model benchmarks. N/A denotes missing standard results for that benchmark.

| Model | Reward-Bench | RM-Bench | LFQA-E |
|---|---|---|---|
| Skywork-Reward-Gemma-2-27B | 75.3% | 69.5% | 52.2% |
| Skywork-Reward-Llama-3.1-8B-v0.2 | 71.8% | 72.6% | 54.0% |
| Gemini-2.5-flash | 77.7% | N/A | 60.6% |
| RM-R1-DeepSeek-Distilled-Qwen-14B | N/A | 71.8% | 59.5% |
| RM-R1-Qwen-Instruct-14B | N/A | 75.6% | 58.4% |

## 5.7 TTRL to Boost Performance

Here, we try to improve the performance on LFQA-E through prompting and reinforcement learning (RL). Firstly, we use a structured prompt to instruct models to embrace their response within a `<answer>...</answer>` tag. This leads to a non-trivial performance increase, considering the relatively lower instruction following ability of small language models. Building on this, since RL is widely used to improve the reasoning ability of LLMs (Cui et al., 2025; Fan et al., 2025), we use RL to leverage the performance of LLMs on LFQA evaluation. Considering the lack of high-quality data, we implement TTRL (Zuo et al., 2025) on our LFQA-E-EN as an example. We configure our model with a batch size of 8, a rollout temperature of 1.0, and generate 32 rollouts per prompt. The learning rate is $5e$-7. During validation, the temperature is 0.0 for consistent results. The reward signal is based on an outcome-based rule, similar to the approach used in Deepseek-R1 (DeepSeek-AI et al., 2025a). As shown in Table 7, TTRL yield a substantial performance boost, demonstrating the effectiveness of using RL for this task. Also, the response length grows steadily during the training until the training rewards converge. However, we observe a rapid convergence where all rollouts produce identical preferences, which limits further improvements. We attribute this to the underlying three-category classification which is easily overfitted when the model is over-confident. For a remedy, we implement clip-higher mechanism used in DAPO (Yu et al., 2025). Also, we observe a performance boost, indicating that when we increase the diversity during rollouts, RL can take effect more stably. However, more sophisticated methods and high-quality data are needed for a better evaluation metric that mimics human preferences.

## 6 Conclusion

We introduce LFQA-E, a multilingual benchmark for LFQA evaluation. It consists of 1625 questions and 7649 comparisons, spanning 15 topics, from natural science to social science, consisting of 3 settings, i.e., *h v. h*, *h v. m*, and *m v. m*. Each records include a clear question, an authorized reference, and two hard-to-differentiate responses, ensuring its difficulty. We conduct experiments on 15 automatic evaluation metrics. The results show that none of the metrics can evaluate long-form responses as well as human beings. We further analyze the generalization of different metrics across languages and settings. The results further indicate that all models struggle to generalize well to all comparisons. We find that LRMs and specifically trained evaluation models lead on LFQA-E. The test-time-scaled evaluation model may be used to enhance the performance of LFQA evaluation.

ETHICS STATEMENT

This paper proposes a benchmark consisting of questions and references from examination papers and online forums. We perform data filtering to ensure that there is no offense in the benchmark data.

REPRODUCIBILITY STATEMENT

We have provide our settings comprehensively for data construction and evaluation so that the result can be reproduced using the same settings and prompt.

ACKNOWLEDGEMENT

We thank the China Construction Third Engineering Bureau Group Co.,Ltd. for their support. w

REFERENCES

Pritom Saha Akash, Kashob Kumar Roy, Lucian Popa, and Kevin Chen-Chuan Chang. Long-form question answering: An iterative planning-retrieval-generation approach, 2023. URL https://arxiv.org/abs/2311.09383.

Yupeng Chang, Xu Wang, Jindong Wang, Yuan Wu, Linyi Yang, Kaijie Zhu, Hao Chen, Xiaoyuan Yi, Cunxiang Wang, Yidong Wang, Wei Ye, Yue Zhang, Yi Chang, Philip S. Yu, Qiang Yang, and Xing Xie. A survey on evaluation of large language models, 2023. URL https://arxiv.org/abs/2307.03109.

Hung-Ting Chen, Fangyuan Xu, Shane Arora, and Eunsol Choi. Understanding retrieval augmentation for long-form question answering, 2023. URL https://arxiv.org/abs/2310.12150.

Xiusi Chen, Gaotang Li, Ziqi Wang, Bowen Jin, Cheng Qian, Yu Wang, Hongru Wang, Yu Zhang, Denghui Zhang, Tong Zhang, Hanghang Tong, and Heng Ji. Rm-r1: Reward modeling as reasoning, 2025. URL https://arxiv.org/abs/2505.02387.

Wei-Lin Chiang, Lianmin Zheng, Ying Sheng, Anastasios Nikolas Angelopoulos, Tianle Li, Dacheng Li, Hao Zhang, Banghua Zhu, Michael Jordan, Joseph E. Gonzalez, and Ion Stoica. Chatbot arena: An open platform for evaluating llms by human preference, 2024.

Ganqu Cui, Lifan Yuan, Zefan Wang, Hanbin Wang, Wendi Li, Bingxiang He, Yuchen Fan, Tianyu Yu, Qixin Xu, Weize Chen, et al. Process reinforcement through implicit rewards. *arXiv preprint arXiv:2502.01456*, 2025.

DeepSeek-AI, Daya Guo, Dejian Yang, Haowei Zhang, Junxiao Song, Ruoyu Zhang, Runxin Xu, Qihao Zhu, Shirong Ma, Peiyi Wang, Xiao Bi, Xiaokang Zhang, Xingkai Yu, Yu Wu, Z. F. Wu, Zhibin Gou, Zhihong Shao, Zhuoshu Li, Ziyi Gao, Aixin Liu, Bing Xue, Bingxuan Wang, Bochao Wu, Bei Feng, Chengda Lu, Chenggang Zhao, Chengqi Deng, Chenyu Zhang, Chong Ruan, Damai Dai, Deli Chen, Dongjie Ji, Erhang Li, Fangyun Lin, Fucong Dai, Fuli Luo, Guangbo Hao, Guanting Chen, Guowei Li, H. Zhang, Han Bao, Hanwei Xu, Haocheng Wang, Honghui Ding, Huajian Xin, Huazuo Gao, Hui Qu, Hui Li, Jianzhong Guo, Jiashi Li, Jiawei Wang, Jingchang Chen, Jingyang Yuan, Junjie Qiu, Junlong Li, J. L. Cai, Jiaqi Ni, Jian Liang, Jin Chen, Kai Dong, Kai Hu, Kaige Gao, Kang Guan, Kexin Huang, Kuai Yu, Lean Wang, Lecong Zhang, Liang Zhao, Litong Wang, Liyue Zhang, Lei Xu, Leyi Xia, Mingchuan Zhang, Minghua Zhang, Minghui Tang, Meng Li, Miaojun Wang, Mingming Li, Ning Tian, Panpan Huang, Peng Zhang, Qiancheng Wang, Qinyu Chen, Qiushi Du, Ruiqi Ge, Ruisong Zhang, Ruizhe Pan, Runji Wang, R. J. Chen, R. L. Jin, Ruyi Chen, Shanghao Lu, Shangyan Zhou, Shanhuang Chen, Shengfeng Ye, Shiyu Wang, Shuiping Yu, Shunfeng Zhou, Shuting Pan, S. S. Li, Shuang Zhou, Shaoqing Wu, Shengfeng Ye, Tao Yun, Tian Pei, Tianyu Sun, T. Wang, Wangding Zeng, Wanjia Zhao, Wen Liu, Wenfeng Liang, Wenjun Gao, Wenqin Yu, Wentao Zhang, W. L. Xiao, Wei An, Xiaodong Liu, Xiaohan Wang, Xiaokang Chen, Xiaotao Nie, Xin Cheng, Xin Liu, Xin Xie, Xingchao Liu,

Xinyu Yang, Xinyuan Li, Xuecheng Su, Xuheng Lin, X. Q. Li, Xiangyue Jin, Xiaojin Shen, Xiaosha Chen, Xiaowen Sun, Xiaoxiang Wang, Xinnan Song, Xinyi Zhou, Xianzu Wang, Xinxia Shan, Y. K. Li, Y. Q. Wang, Y. X. Wei, Yang Zhang, Yanhong Xu, Yao Li, Yao Zhao, Yaofeng Sun, Yaohui Wang, Yi Yu, Yichao Zhang, Yifan Shi, Yiliang Xiong, Ying He, Yishi Piao, Yisong Wang, Yixuan Tan, Yiyang Ma, Yiyuan Liu, Yongqiang Guo, Yuan Ou, Yuduan Wang, Yue Gong, Yuheng Zou, Yujia He, Yunfan Xiong, Yuxiang Luo, Yuxiang You, Yuxuan Liu, Yuyang Zhou, Y. X. Zhu, Yanhong Xu, Yanping Huang, Yaohui Li, Yi Zheng, Yuchen Zhu, Yunxian Ma, Ying Tang, Yukun Zha, Yuting Yan, Z. Z. Ren, Zehui Ren, Zhangli Sha, Zhe Fu, Zhean Xu, Zhenda Xie, Zhengyan Zhang, Zhewen Hao, Zhicheng Ma, Zhigang Yan, Zhiyu Wu, Zihui Gu, Zijia Zhu, Zijun Liu, Zilin Li, Ziwei Xie, Ziyang Song, Zizheng Pan, Zhen Huang, Zhipeng Xu, Zhongyu Zhang, and Zhen Zhang. Deepseek-r1: Incentivizing reasoning capability in llms via reinforcement learning, 2025a. URL https://arxiv.org/abs/2501.12948.

DeepSeek-AI, Aixin Liu, Bei Feng, Bing Xue, Bingxuan Wang, Bochao Wu, Chengda Lu, Chenggang Zhao, Chengqi Deng, Chenyu Zhang, Chong Ruan, Damai Dai, Daya Guo, Dejian Yang, Deli Chen, Dongjie Ji, Erhang Li, Fangyun Lin, Fucong Dai, Fuli Luo, Guangbo Hao, Guanting Chen, Guowei Li, H. Zhang, Han Bao, Hanwei Xu, Haocheng Wang, Haowei Zhang, Honghui Ding, Huajian Xin, Huazuo Gao, Hui Li, Hui Qu, J. L. Cai, Jian Liang, Jianzhong Guo, Jiaqi Ni, Jiashi Li, Jiawei Wang, Jin Chen, Jingchang Chen, Jingyang Yuan, Junjie Qiu, Junlong Li, Junxiao Song, Kai Dong, Kai Hu, Kaige Gao, Kang Guan, Kexin Huang, Kuai Yu, Lean Wang, Lecong Zhang, Lei Xu, Leyi Xia, Liang Zhao, Litong Wang, Liyue Zhang, Meng Li, Miaojun Wang, Mingchuan Zhang, Minghua Zhang, Minghui Tang, Mingming Li, Ning Tian, Panpan Huang, Peiyi Wang, Peng Zhang, Qiancheng Wang, Qihao Zhu, Qinyu Chen, Qiushi Du, R. J. Chen, R. L. Jin, Ruiqi Ge, Ruisong Zhang, Ruizhe Pan, Runji Wang, Runxin Xu, Ruoyu Zhang, Ruyi Chen, S. S. Li, Shanghao Lu, Shangyan Zhou, Shanhuang Chen, Shaoqing Wu, Shengfeng Ye, Shengfeng Ye, Shirong Ma, Shiyu Wang, Shuang Zhou, Shuiping Yu, Shunfeng Zhou, Shuting Pan, T. Wang, Tao Yun, Tian Pei, Tianyu Sun, W. L. Xiao, Wangding Zeng, Wanjia Zhao, Wei An, Wen Liu, Wenfeng Liang, Wenjun Gao, Wenqin Yu, Wentao Zhang, X. Q. Li, Xiangyue Jin, Xianzu Wang, Xiao Bi, Xiaodong Liu, Xiaohan Wang, Xiaojin Shen, Xiaokang Chen, Xiaokang Zhang, Xiaosha Chen, Xiaotao Nie, Xiaowen Sun, Xiaoxiang Wang, Xin Cheng, Xin Liu, Xin Xie, Xingchao Liu, Xingkai Yu, Xinnan Song, Xinxia Shan, Xinyi Zhou, Xinyu Yang, Xinyuan Li, Xuecheng Su, Xuheng Lin, Y. K. Li, Y. Q. Wang, Y. X. Wei, Y. X. Zhu, Yang Zhang, Yanhong Xu, Yanhong Xu, Yanping Huang, Yao Li, Yao Zhao, Yaofeng Sun, Yaohui Li, Yaohui Wang, Yi Yu, Yi Zheng, Yichao Zhang, Yifan Shi, Yiliang Xiong, Ying He, Ying Tang, Yishi Piao, Yisong Wang, Yixuan Tan, Yiyang Ma, Yiyuan Liu, Yongqiang Guo, Yu Wu, Yuan Ou, Yuchen Zhu, Yuduan Wang, Yue Gong, Yuheng Zou, Yujia He, Yukun Zha, Yunfan Xiong, Yunxian Ma, Yuting Yan, Yuxiang Luo, Yuxiang You, Yuxuan Liu, Yuyang Zhou, Z. F. Wu, Z. Z. Ren, Zehui Ren, Zhangli Sha, Zhe Fu, Zhean Xu, Zhen Huang, Zhen Zhang, Zhenda Xie, Zhengyan Zhang, Zhewen Hao, Zhibin Gou, Zhicheng Ma, Zhigang Yan, Zhihong Shao, Zhipeng Xu, Zhiyu Wu, Zhongyu Zhang, Zhuoshu Li, Zihui Gu, Zijia Zhu, Zijun Liu, Zilin Li, Ziwei Xie, Ziyang Song, Ziyi Gao, and Zizheng Pan. Deepseek-v3 technical report, 2025b. URL https://arxiv.org/abs/2412.19437.

Abhimanyu Dubey, Abhinav Jauhri, Abhinav Pandey, Abhishek Kadian, Ahmad Al-Dahle, Aiesha Letman, Akhil Mathur, Alan Schelten, Amy Yang, Angela Fan, Anirudh Goyal, Anthony Hartshorn, Aobo Yang, Archi Mitra, Archie Sravankumar, Artem Korenev, Arthur Hinsvark, Arun Rao, Aston Zhang, Aurelien Rodriguez, Austen Gregerson, Ava Spataru, Baptiste Roziere, Bethany Biron, Binh Tang, Bobbie Chern, Charlotte Caucheteux, Chaya Nayak, Chloe Bi, Chris Marra, Chris McConnell, Christian Keller, Christophe Touret, Chunyang Wu, Corinne Wong, Cristian Canton Ferrer, Cyrus Nikolaidis, Damien Allonsius, Daniel Song, Danielle Pintz, Danny Livshits, David Esiobu, Dhruv Choudhary, Dhruv Mahajan, Diego Garcia-Olano, Diego Perino, Dieuwke Hupkes, Egor Lakomkin, Ehab AlBadawy, Elina Lobanova, Emily Dinan, Eric Michael Smith, Filip Radenovic, Frank Zhang, Gabriel Synnaeve, Gabrielle Lee, Georgia Lewis Anderson, Graeme Nail, Gregoire Mialon, Guan Pang, Guillem Cucurell, Hailey Nguyen, Hannah Korevaar, Hu Xu, Hugo Touvron, Iliyan Zarov, Imanol Arrieta Ibarra, Isabel Kloumann, Ishan Misra, Ivan Evtimov, Jade Copet, Jaewon Lee, Jan Geffert, Jana Vranes, Jason Park, Jay Mahadeokar, Jeet Shah, Jelmer van der Linde, Jennifer Billock, Jenny Hong, Jenya Lee, Jeremy Fu, Jianfeng Chi, Jianyu Huang, Jiawen Liu, Jie Wang, Jiecao Yu, Joanna Bitton, Joe Spisak, Jongsoo Park, Joseph Rocca, Joshua Johnstun, Joshua Saxe, Junteng Jia, Kalyan Vasuden Alwala, Kartikeya Upasani, Kate Plawiak, Ke Li, Kenneth Heafield, Kevin Stone, Khalid El-Arini,

Krithika Iyer, Kshitiz Malik, Kuenley Chiu, Kunal Bhalla, Lauren Rantala-Yeary, Laurens van der Maaten, Lawrence Chen, Liang Tan, Liz Jenkins, Louis Martin, Lovish Madaan, Lubo Malo, Lukas Blecher, Lukas Landzaat, Luke de Oliveira, Madeline Muzzi, Mahesh Pasupuleti, Mannat Singh, Manohar Paluri, Marcin Kardas, Mathew Oldham, Mathieu Rita, Maya Pavlova, Melanie Kambadur, Mike Lewis, Min Si, Mitesh Kumar Singh, Mona Hassan, Naman Goyal, Narjes Torabi, Nikolay Bashlykov, Nikolay Bogoychev, Niladri Chatterji, Olivier Duchenne, Onur Çelebi, Patrick Alrassy, Pengchuan Zhang, Pengwei Li, Petar Vasic, Peter Weng, Prajjwal Bhargava, Pratik Dubal, Praveen Krishnan, Punit Singh Koura, Puxin Xu, Qing He, Qingxiao Dong, Ragavan Srinivasan, Raj Ganapathy, Ramon Calderer, Ricardo Silveira Cabral, Robert Stojnic, Roberta Raileanu, Rohit Girdhar, Rohit Patel, Romain Sauvestre, Ronnie Polidoro, Roshan Sumbaly, Ross Taylor, Ruan Silva, Rui Hou, Rui Wang, Saghar Hosseini, Sahana Chennabasappa, Sanjay Singh, Sean Bell, Seohyun Sonia Kim, Sergey Edunov, Shaoliang Nie, Sharan Narang, Sharath Raparthy, Sheng Shen, Shengye Wan, Shruti Bhosale, Shun Zhang, Simon Vandenhende, Soumya Batra, Spencer Whitman, Sten Sootla, Stephane Collot, Suchin Gururangan, Sydney Borodinsky, Tamar Herman, Tara Fowler, Tarek Sheasha, Thomas Georgiou, Thomas Scialom, Tobias Speckbacher, Todor Mihaylov, Tong Xiao, Ujjwal Karn, Vedanuj Goswami, Vibhor Gupta, Vignesh Ramanathan, Viktor Kerkez, Vincent Gonguet, Virginie Do, Vish Vogeti, Vladan Petrovic, Weiwei Chu, Wenhan Xiong, Wenyin Fu, Whitney Meers, Xavier Martinet, Xiaodong Wang, Xiaoqing Ellen Tan, Xinfeng Xie, Xuchao Jia, Xuewei Wang, Yaelle Goldschlag, Yashesh Gaur, Yasmine Babaei, Yi Wen, Yiwen Song, Yuchen Zhang, Yue Li, Yuning Mao, Zacharie Delpierre Coudert, Zheng Yan, Zhengxing Chen, Zoe Papakipos, Aaditya Singh, Aaron Grattafiori, Abha Jain, Adam Kelsey, Adam Shajnfeld, Adithya Gangidi, Adolfo Victoria, Ahuva Goldstand, Ajay Menon, Ajay Sharma, Alex Boesenberg, Alex Vaughan, Alexei Baevski, Allie Feinstein, Amanda Kallet, Amit Sangani, Anam Yunus, Andrei Lupu, Andres Alvarado, Andrew Caples, Andrew Gu, Andrew Ho, Andrew Poulton, Andrew Ryan, Ankit Ramchandani, Annie Franco, Aparajita Saraf, Arkabandhu Chowdhury, Ashley Gabriel, Ashwin Bharambe, Assaf Eisenman, Azadeh Yazdan, Beau James, Ben Maurer, Benjamin Leonhardi, Bernie Huang, Beth Loyd, Beto De Paola, Bhargavi Paranjape, Bing Liu, Bo Wu, Boyu Ni, Braden Hancock, Bram Wasti, Brandon Spence, Brani Stojkovic, Brian Gamido, Britt Montalvo, Carl Parker, Carly Burton, Catalina Mejia, Changhan Wang, Changkyu Kim, Chao Zhou, Chester Hu, Ching-Hsiang Chu, Chris Cai, Chris Tindal, Christoph Feichtenhofer, Damon Civin, Dana Beaty, Daniel Kreymer, Daniel Li, Danny Wyatt, David Adkins, David Xu, Davide Testuggine, Delia David, Devi Parikh, Diana Liskovich, Didem Foss, Dingkang Wang, Duc Le, Dustin Holland, Edward Dowling, Eissa Jamil, Elaine Montgomery, Eleonora Presani, Emily Hahn, Emily Wood, Erik Brinkman, Esteban Arcaute, Evan Dunbar, Evan Smothers, Fei Sun, Felix Kreuk, Feng Tian, Firat Ozgenel, Francesco Caggioni, Francisco Guzmán, Frank Kanayet, Frank Seide, Gabriela Medina Florez, Gabriella Schwarz, Gada Badeer, Georgia Swee, Gil Halpern, Govind Thattai, Grant Herman, Grigory Sizov, Guangyi, Zhang, Guna Lakshminarayanan, Hamid Shojanazeri, Han Zou, Hannah Wang, Hanwen Zha, Haroun Habeeb, Harrison Rudolph, Helen Suk, Henry Aspegren, Hunter Goldman, Igor Molybog, Igor Tufanov, Irina-Elena Veliche, Itai Gat, Jake Weissman, James Geboski, James Kohli, Japhet Asher, Jean-Baptiste Gaya, Jeff Marcus, Jeff Tang, Jennifer Chan, Jenny Zhen, Jeremy Reizenstein, Jeremy Teboul, Jessica Zhong, Jian Jin, Jingyi Yang, Joe Cummings, Jon Carvill, Jon Shepard, Jonathan McPhie, Jonathan Torres, Josh Ginsburg, Junjie Wang, Kai Wu, Kam Hou U, Karan Saxena, Karthik Prasad, Kartikay Khandelwal, Katayoun Zand, Kathy Matosich, Kaushik Veeraraghavan, Kelly Michelena, Keqian Li, Kun Huang, Kunal Chawla, Kushal Lakhotia, Kyle Huang, Lailin Chen, Lakshya Garg, Lavender A, Leandro Silva, Lee Bell, Lei Zhang, Liangpeng Guo, Licheng Yu, Liron Moshkovich, Luca Wehrstedt, Madian Khabsa, Manav Avalani, Manish Bhatt, Maria Tsimpoukelli, Martynas Mankus, Matan Hasson, Matthew Lennie, Matthias Reso, Maxim Groshev, Maxim Naumov, Maya Lathi, Meghan Keneally, Michael L. Seltzer, Michal Valko, Michelle Restrepo, Mihir Patel, Mik Vyatskov, Mikayel Samvelyan, Mike Clark, Mike Macey, Mike Wang, Miquel Jubert Hermoso, Mo Metanat, Mohammad Rastegari, Munish Bansal, Nandhini Santhanam, Natascha Parks, Natasha White, Navyata Bawa, Nayan Singhal, Nick Egebo, Nicolas Usunier, Nikolay Pavlovich Laptev, Ning Dong, Ning Zhang, Norman Cheng, Oleg Chernoguz, Olivia Hart, Omkar Salpekar, Ozlem Kalinli, Parkin Kent, Parth Parekh, Paul Saab, Pavan Balaji, Pedro Rittner, Philip Bontrager, Pierre Roux, Piotr Dollar, Polina Zvyagina, Prashant Ratanchandani, Pritish Yuvraj, Qian Liang, Rachad Alao, Rachel Rodriguez, Rafi Ayub, Raghotham Murthy, Raghu Nayani, Rahul Mitra, Raymond Li, Rebekkah Hogan, Robin Battey, Rocky Wang, Rohan Maheswari, Russ Howes, Ruty Rinott, Sai Jayesh Bondu, Samyak Datta, Sara Chugh, Sara Hunt, Sargun Dhillon, Sasha Sidorov, Sa-

tadru Pan, Saurabh Verma, Seiji Yamamoto, Sharadh Ramaswamy, Shaun Lindsay, Shaun Lindsay, Sheng Feng, Shenghao Lin, Shengxin Cindy Zha, Shiva Shankar, Shuqiang Zhang, Shuqiang Zhang, Sinong Wang, Sneha Agarwal, Soji Sajuyigbe, Soumith Chintala, Stephanie Max, Stephen Chen, Steve Kehoe, Steve Satterfield, Sudarshan Govindaprasad, Sumit Gupta, Sungmin Cho, Sunny Virk, Suraj Subramanian, Sy Choudhury, Sydney Goldman, Tal Remez, Tamar Glaser, Tamara Best, Thilo Kohler, Thomas Robinson, Tianhe Li, Tianjun Zhang, Tim Matthews, Timothy Chou, Tzook Shaked, Varun Vontimitta, Victoria Ajayi, Victoria Montanez, Vijai Mohan, Vinay Satish Kumar, Vishal Mangla, Vlad Ionescu, Vlad Poenaru, Vlad Tiberiu Mihailescu, Vladimir Ivanov, Wei Li, Wenchen Wang, Wenwen Jiang, Wes Bouaziz, Will Constable, Xiaocheng Tang, Xiaofang Wang, Xiaojian Wu, Xiaolan Wang, Xide Xia, Xilun Wu, Xinbo Gao, Yanjun Chen, Ye Hu, Ye Jia, Ye Qi, Yenda Li, Yilin Zhang, Ying Zhang, Yossi Adi, Youngjin Nam, Yu, Wang, Yuchen Hao, Yundi Qian, Yuzi He, Zach Rait, Zachary DeVito, Zef Rosnbrick, Zhaoduo Wen, Zhenyu Yang, and Zhiwei Zhao. The llama 3 herd of models, 2024. URL https://arxiv.org/abs/2407.21783.

Angela Fan, Yacine Jernite, Ethan Perez, David Grangier, Jason Weston, and Michael Auli. Eli5: Long form question answering, 2019. URL https://arxiv.org/abs/1907.09190.

Yuchen Fan, Yantao Liu, Zijun Yao, Jifan Yu, Lei Hou, and Juanzi Li. Evaluating generative language models in information extraction as subjective question correction. *arXiv preprint arXiv:2404.03532*, 2024a.

Yuchen Fan, Xin Zhong, Yazhe Wan, Chengsi Wang, Haonan Cheng, Gaoche Wu, Ning Ding, and Bowen Zhou. Eva-score: Evaluating abstractive long-form summarization on informativeness through extraction and validation. *arXiv preprint arXiv:2407.04969*, 2024b.

Yuchen Fan, Kaiyan Zhang, Heng Zhou, Yuxin Zuo, Yanxu Chen, Yu Fu, Xinwei Long, Xuekai Zhu, Che Jiang, Yuchen Zhang, et al. Ssrl: Self-search reinforcement learning. *arXiv preprint arXiv:2508.10874*, 2025.

Dongfu Jiang, Yishan Li, Ge Zhang, Wenhao Huang, Bill Yuchen Lin, and Wenhu Chen. Tigerscore: Towards building explainable metric for all text generation tasks, 2024. URL https://arxiv.org/abs/2310.00752.

Mandar Joshi, Eunsol Choi, Daniel S. Weld, and Luke Zettlemoyer. Triviaqa: A large scale distantly supervised challenge dataset for reading comprehension, 2017. URL https://arxiv.org/abs/1705.03551.

Seungone Kim, Juyoung Suk, Shayne Longpre, Bill Yuchen Lin, Jamin Shin, Sean Welleck, Graham Neubig, Moontae Lee, Kyungjae Lee, and Minjoon Seo. Prometheus 2: An open source language model specialized in evaluating other language models, 2024.

Tomáš Kočiský, Jonathan Schwarz, Phil Blunsom, Chris Dyer, Karl Moritz Hermann, Gábor Melis, and Edward Grefenstette. The narrativeqa reading comprehension challenge, 2017. URL https://arxiv.org/abs/1712.07040.

Kalpesh Krishna, Aurko Roy, and Mohit Iyyer. Hurdles to progress in long-form question answering, 2021. URL https://arxiv.org/abs/2103.06332.

Nathan Lambert, Valentina Pyatkin, Jacob Morrison, LJ Miranda, Bill Yuchen Lin, Khyathi Chandu, Nouha Dziri, Sachin Kumar, Tom Zick, Yejin Choi, Noah A. Smith, and Hannaneh Hajishirzi. Rewardbench: Evaluating reward models for language modeling. https://huggingface.co/spaces/allenai/reward-bench, 2024.

Junlong Li, Shichao Sun, Weizhe Yuan, Run-Ze Fan, Hai Zhao, and Pengfei Liu. Generative judge for evaluating alignment, 2023. URL https://arxiv.org/abs/2310.05470.

Chin-Yew Lin. ROUGE: A package for automatic evaluation of summaries. In *Text Summarization Branches Out*, pp. 74–81, Barcelona, Spain, July 2004. Association for Computational Linguistics. URL https://aclanthology.org/W04-1013.

Chris Yuhao Liu, Liang Zeng, Jiacai Liu, Rui Yan, Jujie He, Chaojie Wang, Shuicheng Yan, Yang Liu, and Yahui Zhou. Skywork-reward: Bag of tricks for reward modeling in llms. *arXiv preprint arXiv:2410.18451*, 2024a.

Xiusi Liu, Gaotang Li, Ziqi Wang, Bowen Jin, Cheng Qian, Yu Wang, Hongru Wang, Yu Zhang, Denghui Zhang, Tong Zhang, Hanghang Tong, and Heng Ji. Rm-r1: Reward modeling as reasoning, 2024b. URL `https://arxiv.org/abs/2505.02387`.

Yang Liu, Dan Iter, Yichong Xu, Shuohang Wang, Ruochen Xu, and Chenguang Zhu. G-eval: Nlg evaluation using gpt-4 with better human alignment, 2023. URL `https://arxiv.org/abs/2303.16634`.

Yantao Liu, Zijun Yao, Rui Min, Yixin Cao, Lei Hou, and Juanzi Li. Rm-bench: Benchmarking reward models of language models with subtlety and style, 2024c. URL `https://arxiv.org/abs/2410.16184`.

Sewon Min, Kalpesh Krishna, Xinxi Lyu, Mike Lewis, Wen tau Yih, Pang Wei Koh, Mohit Iyyer, Luke Zettlemoyer, and Hannaneh Hajishirzi. Factscore: Fine-grained atomic evaluation of factual precision in long form text generation, 2023. URL `https://arxiv.org/abs/2305.14251`.

Reiichiro Nakano, Jacob Hilton, Suchir Balaji, Jeff Wu, Long Ouyang, Christina Kim, Christopher Hesse, Shantanu Jain, Vineet Kosaraju, William Saunders, Xu Jiang, Karl Cobbe, Tyna Eloundou, Gretchen Krueger, Kevin Button, Matthew Knight, Benjamin Chess, and John Schulman. Webgpt: Browser-assisted question-answering with human feedback, 2022. URL `https://arxiv.org/abs/2112.09332`.

OpenAI. Chatgpt: Chat generative pre-trained transformer. `https://chat.openai.com/`, 2023. Accessed: 2024-08-05.

OpenAI. Hello gpt-4o. `https://openai.com/index/hello-gpt-4o/`, 2024. Accessed: 2024-08-05.

OpenAI, :, Aaron Hurst, Adam Lerer, Adam P. Goucher, Adam Perelman, Aditya Ramesh, Aidan Clark, AJ Ostrow, Akila Welihinda, Alan Hayes, Alec Radford, Aleksander Mądry, Alex Baker-Whitcomb, Alex Beutel, Alex Borzunov, Alex Carney, Alex Chow, Alex Kirillov, Alex Nichol, Alex Paino, Alex Renzin, Alex Tachard Passos, Alexander Kirillov, Alexi Christakis, Alexis Conneau, Ali Kamali, Allan Jabri, Allison Moyer, Allison Tam, Amadou Crookes, Amin Tootoochian, Amin Tootoonchian, Ananya Kumar, Andrea Vallone, Andrej Karpathy, Andrew Braunstein, Andrew Cann, Andrew Codispoti, Andrew Galu, Andrew Kondrich, Andrew Tulloch, Andrey Mishchenko, Angela Baek, Angela Jiang, Antoine Pelisse, Antonia Woodford, Anuj Gosalia, Arka Dhar, Ashley Pantuliano, Avi Nayak, Avital Oliver, Barret Zoph, Behrooz Ghorbani, Ben Leimberger, Ben Rossen, Ben Sokolowsky, Ben Wang, Benjamin Zweig, Beth Hoover, Blake Samic, Bob McGrew, Bobby Spero, Bogo Giertler, Bowen Cheng, Brad Lightcap, Brandon Walkin, Brendan Quinn, Brian Guarraci, Brian Hsu, Bright Kellogg, Brydon Eastman, Camillo Lugaresi, Carroll Wainwright, Cary Bassin, Cary Hudson, Casey Chu, Chad Nelson, Chak Li, Chan Jun Shern, Channing Conger, Charlotte Barette, Chelsea Voss, Chen Ding, Cheng Lu, Chong Zhang, Chris Beaumont, Chris Hallacy, Chris Koch, Christian Gibson, Christina Kim, Christine Choi, Christine McLeavey, Christopher Hesse, Claudia Fischer, Clemens Winter, Coley Czarnecki, Colin Jarvis, Colin Wei, Constantin Koumouzelis, Dane Sherburn, Daniel Kappler, Daniel Levin, Daniel Levy, David Carr, David Farhi, David Mely, David Robinson, David Sasaki, Denny Jin, Dev Valladares, Dimitris Tsipras, Doug Li, Duc Phong Nguyen, Duncan Findlay, Edede Oiwoh, Edmund Wong, Ehsan Asdar, Elizabeth Proehl, Elizabeth Yang, Eric Antonow, Eric Kramer, Eric Peterson, Eric Sigler, Eric Wallace, Eugene Brevdo, Evan Mays, Farzad Khorasani, Felipe Petroski Such, Filippo Raso, Francis Zhang, Fred von Lohmann, Freddie Sulit, Gabriel Goh, Gene Oden, Geoff Salmon, Giulio Starace, Greg Brockman, Hadi Salman, Haiming Bao, Haitang Hu, Hannah Wong, Haoyu Wang, Heather Schmidt, Heather Whitney, Heewoo Jun, Hendrik Kirchner, Henrique Ponde de Oliveira Pinto, Hongyu Ren, Huiwen Chang, Hyung Won Chung, Ian Kivlichan, Ian O'Connell, Ian O'Connell, Ian Osband, Ian Silber, Ian Sohl, Ibrahim Okuyucu, Ikai Lan, Ilya Kostrikov, Ilya Sutskever, Ingmar Kanitscheider, Ishaan Gulrajani, Jacob Coxon, Jacob Menick, Jakub Pachocki, James Aung, James Betker, James Crooks, James Lennon, Jamie Kiros, Jan Leike, Jane Park, Jason Kwon, Jason Phang, Jason Teplitz, Jason Wei, Jason Wolfe, Jay Chen, Jeff Harris, Jenia Varavva, Jessica Gan Lee, Jessica Shieh, Ji Lin, Jiahui Yu, Jiayi Weng, Jie Tang, Jieqi Yu, Joanne Jang, Joaquin Quinonero Candela, Joe Beutler, Joe Landers, Joel Parish, Johannes Heidecke, John Schulman, Jonathan Lachman, Jonathan McKay,

Jonathan Uesato, Jonathan Ward, Jong Wook Kim, Joost Huizinga, Jordan Sitkin, Jos Kraaijeveld, Josh Gross, Josh Kaplan, Josh Snyder, Joshua Achiam, Joy Jiao, Joyce Lee, Juntang Zhuang, Justyn Harriman, Kai Fricke, Kai Hayashi, Karan Singhal, Katy Shi, Kavin Karthik, Kayla Wood, Kendra Rimbach, Kenny Hsu, Kenny Nguyen, Keren Gu-Lemberg, Kevin Button, Kevin Liu, Kiel Howe, Krithika Muthukumar, Kyle Luther, Lama Ahmad, Larry Kai, Lauren Itow, Lauren Workman, Leher Pathak, Leo Chen, Li Jing, Lia Guy, Liam Fedus, Liang Zhou, Lien Mamitsuka, Lilian Weng, Lindsay McCallum, Lindsey Held, Long Ouyang, Louis Feuvrier, Lu Zhang, Lukas Kondraciuk, Lukasz Kaiser, Luke Hewitt, Luke Metz, Lyric Doshi, Mada Aflak, Maddie Simens, Madelaine Boyd, Madeleine Thompson, Marat Dukhan, Mark Chen, Mark Gray, Mark Hudnall, Marvin Zhang, Marwan Aljubeh, Mateusz Litwin, Matthew Zeng, Max Johnson, Maya Shetty, Mayank Gupta, Meghan Shah, Mehmet Yatbaz, Meng Jia Yang, Mengchao Zhong, Mia Glaese, Mianna Chen, Michael Janner, Michael Lampe, Michael Petrov, Michael Wu, Michele Wang, Michelle Fradin, Michelle Pokrass, Miguel Castro, Miguel Oom Temudo de Castro, Mikhail Pavlov, Miles Brundage, Miles Wang, Minal Khan, Mira Murati, Mo Bavarian, Molly Lin, Murat Yesildal, Nacho Soto, Natalia Gimelshein, Natalie Cone, Natalie Staudacher, Natalie Summers, Natan LaFontaine, Neil Chowdhury, Nick Ryder, Nick Stathas, Nick Turley, Nik Tezak, Niko Felix, Nithanth Kudige, Nitish Keskar, Noah Deutsch, Noel Bundick, Nora Puckett, Ofir Nachum, Ola Okelola, Oleg Boiko, Oleg Murk, Oliver Jaffe, Olivia Watkins, Olivier Godement, Owen Campbell-Moore, Patrick Chao, Paul McMillan, Pavel Belov, Peng Su, Peter Bak, Peter Bakkum, Peter Deng, Peter Dolan, Peter Hoeschele, Peter Welinder, Phil Tillet, Philip Pronin, Philippe Tillet, Prafulla Dhariwal, Qiming Yuan, Rachel Dias, Rachel Lim, Rahul Arora, Rajan Troll, Randall Lin, Rapha Gontijo Lopes, Raul Puri, Reah Miyara, Reimar Leike, Renaud Gaubert, Reza Zamani, Ricky Wang, Rob Donnelly, Rob Honsby, Rocky Smith, Rohan Sahai, Rohit Ramchandani, Romain Huet, Rory Carmichael, Rowan Zellers, Roy Chen, Ruby Chen, Ruslan Nigmatullin, Ryan Cheu, Saachi Jain, Sam Altman, Sam Schoenholz, Sam Toizer, Samuel Miserendino, Sandhini Agarwal, Sara Culver, Scott Ethersmith, Scott Gray, Sean Grove, Sean Metzger, Shamez Hermani, Shantanu Jain, Shengjia Zhao, Sherwin Wu, Shino Jomoto, Shirong Wu, Shuaiqi, Xia, Sonia Phene, Spencer Papay, Srinivas Narayanan, Steve Coffey, Steve Lee, Stewart Hall, Suchir Balaji, Tal Broda, Tal Stramer, Tao Xu, Tarun Gogineni, Taya Christianson, Ted Sanders, Tejal Patwardhan, Thomas Cunninghman, Thomas Degry, Thomas Dimson, Thomas Raoux, Thomas Shadwell, Tianhao Zheng, Todd Underwood, Todor Markov, Toki Sherbakov, Tom Rubin, Tom Stasi, Tomer Kaftan, Tristan Heywood, Troy Peterson, Tyce Walters, Tyna Eloundou, Valerie Qi, Veit Moeller, Vinnie Monaco, Vishal Kuo, Vlad Fomenko, Wayne Chang, Weiyi Zheng, Wenda Zhou, Wesam Manassra, Will Sheu, Wojciech Zaremba, Yash Patil, Yilei Qian, Yongjik Kim, Youlong Cheng, Yu Zhang, Yuchen He, Yuchen Zhang, Yujia Jin, Yunxing Dai, and Yury Malkov. Gpt-4o system card, 2024. URL https://arxiv.org/abs/2410.21276.

Qwen, :, An Yang, Baosong Yang, Beichen Zhang, Binyuan Hui, Bo Zheng, Bowen Yu, Chengyuan Li, Dayiheng Liu, Fei Huang, Haoran Wei, Huan Lin, Jian Yang, Jianhong Tu, Jianwei Zhang, Jianxin Yang, Jiaxi Yang, Jingren Zhou, Junyang Lin, Kai Dang, Keming Lu, Keqin Bao, Kexin Yang, Le Yu, Mei Li, Mingfeng Xue, Pei Zhang, Qin Zhu, Rui Men, Runji Lin, Tianhao Li, Tianyi Tang, Tingyu Xia, Xingzhang Ren, Xuancheng Ren, Yang Fan, Yang Su, Yichang Zhang, Yu Wan, Yuqiong Liu, Zeyu Cui, Zhenru Zhang, and Zihan Qiu. Qwen2.5 technical report, 2025. URL https://arxiv.org/abs/2412.15115.

Pranav Rajpurkar, Jian Zhang, Konstantin Lopyrev, and Percy Liang. Squad: 100,000+ questions for machine comprehension of text, 2016. URL https://arxiv.org/abs/1606.05250.

Jason Wei, Xuezhi Wang, Dale Schuurmans, Maarten Bosma, Brian Ichter, Fei Xia, Ed Chi, Quoc Le, and Denny Zhou. Chain-of-thought prompting elicits reasoning in large language models, 2023. URL https://arxiv.org/abs/2201.11903.

Fangyuan Xu, Junyi Jessy Li, and Eunsol Choi. How do we answer complex questions: Discourse structure of long-form answers, 2022. URL https://arxiv.org/abs/2203.11048.

Fangyuan Xu, Yixiao Song, Mohit Iyyer, and Eunsol Choi. A critical evaluation of evaluations for long-form question answering, 2023. URL https://arxiv.org/abs/2305.18201.

Ruijie Xu, Zengzhi Wang, Run-Ze Fan, and Pengfei Liu. Benchmarking benchmark leakage in large language models. *arXiv preprint arXiv:2404.18824*, 2024. URL https://arxiv.org/abs/2404.18824.

Qiying Yu, Zheng Zhang, Ruofei Zhu, Yufeng Yuan, Xiaochen Zuo, Yu Yue, Weinan Dai, Tiantian Fan, Gaohong Liu, Lingjun Liu, Xin Liu, Haibin Lin, Zhiqi Lin, Bole Ma, Guangming Sheng, Yuxuan Tong, Chi Zhang, Mofan Zhang, Wang Zhang, Hang Zhu, Jinhua Zhu, Jiaze Chen, Jiangjie Chen, Chengyi Wang, Hongli Yu, Yuxuan Song, Xiangpeng Wei, Hao Zhou, Jingjing Liu, Wei-Ying Ma, Ya-Qin Zhang, Lin Yan, Mu Qiao, Yonghui Wu, and Mingxuan Wang. Dapo: An open-source llm reinforcement learning system at scale, 2025. URL https://arxiv.org/abs/2503.14476.

Weizhe Yuan, Graham Neubig, and Pengfei Liu. Bartscore: Evaluating generated text as text generation, 2021. URL https://arxiv.org/abs/2106.11520.

Kaiyan Zhang, Yuxin Zuo, Bingxiang He, Youbang Sun, Runze Liu, Che Jiang, Yuchen Fan, Kai Tian, Guoli Jia, Pengfei Li, et al. A survey of reinforcement learning for large reasoning models. *arXiv preprint arXiv:2509.08827*, 2025.

Tianyi Zhang, Varsha Kishore, Felix Wu, Kilian Q. Weinberger, and Yoav Artzi. Bertscore: Evaluating text generation with bert, 2020. URL https://arxiv.org/abs/1904.09675.

Lianmin Zheng, Wei-Lin Chiang, Ying Sheng, Siyuan Zhuang, Zhanghao Wu, Yonghao Zhuang, Zi Lin, Zhuohan Li, Dacheng Li, Eric Xing, Hao Zhang, Joseph E. Gonzalez, and Ion Stoica. Judging llm-as-a-judge with mt-bench and chatbot arena. In *Thirty-seventh Conference on Neural Information Processing Systems Datasets and Benchmarks Track*, 2023. URL https://openreview.net/forum?id=uccHPGDlao.

Lianmin Zheng, Wei-Lin Chiang, Ying Sheng, Tianle Li, Siyuan Zhuang, Zhanghao Wu, Yonghao Zhuang, Zhuohan Li, Zi Lin, Eric Xing, Joseph E. Gonzalez, Ion Stoica, and Hao Zhang. Lmsyschat-1m: A large-scale real-world llm conversation dataset. In *The Twelfth International Conference on Learning Representations*, 2024. URL https://openreview.net/forum?id=BOfDKxfwt0.

Heng Zhou, Hejia Geng, Xiangyuan Xue, Li Kang, Yiran Qin, Zhiyong Wang, Zhenfei Yin, and Lei Bai. Reso: A reward-driven self-organizing llm-based multi-agent system for reasoning tasks. *arXiv preprint arXiv:2503.02390*, 2025.

Yuxin Zuo, Kaiyan Zhang, Li Sheng, Shang Qu, Ganqu Cui, Xuekai Zhu, Haozhan Li, Yuchen Zhang, Xinwei Long, Ermo Hua, et al. Ttrl: Test-time reinforcement learning. *arXiv preprint arXiv:2504.16084*, 2025.

# A    USE OF LLMS

We use LLMs to refine our writing using Gemini2.5-Pro, GPT-5, and Claude-4.1. We check the refined phrases after generation.

# B    ADDITIONAL RESULTS

## B.1    ANNOTATION OF LLM PARAPHRASING

We randomly sample 100 records to compare the original responses and the paraphrased responses. We define three types of errors: 1) The paraphrased response contains factual errors; 2) The paraphrased response adds new core points to answer the problem, which may reverse the order; 3) The paraphrased response drops original points or introduces influency or inconsistency. After annotation, we find *error 2* in only 1 out of 100 paraphrased response, indicating the effectiveness of using GPT-4o to paraphrase responses.

## B.2    PROMPT SENSITIVITY ANALYSIS

To further investigate the sensitivity to prompt design, we conduct an additional ablation by instructing models to solely provide a final answer without CoT. The results are presented in the table below. This experiment further highlights the importance of CoT for enhancing the ability when evaluating: for most settings under LFQA-E-EN, the performance degrades greatly. While for LFQA-E-ZH, the performance is hard to predict, indicating the language vulnerability.

Table 10: Ablation study on prompt design.

| Model | LFQA-E-EN | | LFQA-E-ZH | |
| --- | --- | --- | --- | --- |
| | Acc | F1 | Acc | F1 |
| Qwen2.5-32B-Instruct | $57.3_{\downarrow 6.2}$ | $45.5_{\downarrow 0.3}$ | $52.8_{\downarrow 3.9}$ | $38.6_{\downarrow 3.2}$ |
| Qwen2.5-72B-Instruct | $60.4_{\downarrow 0.8}$ | $45.1_{\uparrow 2.0}$ | $53.9_{\uparrow 0.9}$ | $37.0_{\downarrow 2.0}$ |
| GPT-4o | $61.4_{\downarrow 0.3}$ | $44.9_{\downarrow 1.5}$ | $51.5_{\downarrow 1.7}$ | $37.7_{\downarrow 4.9}$ |
| DeepSeek-V3 | $51.9_{\downarrow 6.0}$ | $42.2_{\uparrow 2.9}$ | $54.4_{\uparrow 0.6}$ | $37.0_{\downarrow 4.1}$ |
| DeepSeek-R1 | $54.8_{\downarrow 4.8}$ | $41.4_{\downarrow 1.5}$ | $60.0_{\uparrow 2.2}$ | $44.1_{\uparrow 1.7}$ |

## B.3    THE ABILITY OF EVALUATION METRICS VARIES ACROSS DOMAINS.

The performance of evaluation metrics in LFQA-E varies significantly across different domains. As shown in Tables 11 and 12, models and metrics exhibit distinct strengths and weaknesses depending on the subject area. For instance, on the LFQA-E-ZH dataset, models like Qwen2.5-32B-Instruct and GPT-4o consistently excel in subjects such as Geography, Law, and Medicine but perform less effectively in more complex domains like Psychology and History. Conversely, models like DeepSeek-V3 and RM-based Evaluation Metrics show particular strengths in fields like Politics and Law, where the emphasis is on factual accuracy and legal context. Similarly, in the LFQA-E-EN dataset, LLM-based models such as Qwen2.5-32B-Instruct perform exceptionally well in Engineering and Technology but show a noticeable decline in subjects like Psychology and Mathematics, possibly due to the more abstract nature of these domains. In contrast, RM-based Evaluation Metrics, such as RM-R1-DeepSeek-Distilled-Qwen-14B, demonstrate impressive performance in fields like Planetary Science and Chemistry, where the data is often more structured and less ambiguous.

## B.4    RESULTS OF ADDITIONAL METRICS

We incorporate several additional baselines, including factuality-oriented and evaluation-ensemble measures, as shown below in Table 13. Through the experimental results, we find that the agentic

Table 11: Results of different topics from LFQA-E-ZH. The largest value is denoted using **bold**.

| METRICS | GEOGRAPHY | HISTORY | POLITICS | PSYCHOLOGY | MEDICINE | LAW |
|---|---|---|---|---|---|---|
| LLM-based Evaluation Metric | | | | | | |
| Qwen2.5-32B-Instruct | 64.3 | 58.2 | 50.4 | 44.8 | 65.1 | 49.0 |
| Qwen2.5-72B-Instruct | **65.5** | 54.3 | 51.1 | 42.7 | 58.3 | 54.9 |
| Llama-3.1-70B-Instruct | 41.5 | 34.4 | 37.0 | 46.3 | 46.6 | **62.6** |
| GPT-4o | 62.0 | 51.8 | 49.1 | 44.8 | 58.3 | 46.1 |
| Deepseek-V3 | 60.0 | 50.5 | 54.4 | 40.6 | 57.3 | 49.0 |
| RM-based Evaluation Metric | | | | | | |
| Skywork-Reward-Llama | 58.3 | 54.1 | 51.4 | 34.4 | 59.2 | 54.9 |
| Skywork-Reward-Gemma | 54.3 | 50.0 | 41.0 | 36.5 | 48.5 | 56.9 |
| RM-R1-Qwen2.5-Instruct-14B | 58.6 | 51.0 | 45.5 | 44.8 | 58.4 | 54.9 |
| RM-R1-DeepSeek-Distilled-Qwen-14B | 61.4 | 51.8 | **62.5** | 37.5 | 54.5 | 57.8 |
| LRM-based Evaluation Metric | | | | | | |
| o1-mini | 53.5 | 46.9 | 42.5 | **49.0** | **66.0** | 56.9 |
| Deepseek-R1 | 64.0 | **58.4** | 53.2 | 42.7 | 67.0 | 54.9 |
| Trained Evaluation Metric | | | | | | |
| Auto-J-6B-bilingual | 58.5 | 55.4 | 41.7 | 38.5 | 58.4 | 59.8 |
| Promethus-7B-v2.0 | 54.0 | 52.6 | 43.2 | 40.6 | 51.5 | 59.8 |
| M-Promethus-14B | 56.0 | 51.3 | 43.7 | 41.7 | 44.7 | 50.0 |

Table 12: Results of different topics from LFQA-E-EN. The largest value is denoted using **bold**.

| METRICS | ENGINEERING | BIOLOGY | TECHNOLOGY | PHYSICS | MATHEMATICS | ECONOMICS | PLANETARY SCIENCE | CHEMISTRY | OTHER |
|---|---|---|---|---|---|---|---|---|---|
| LLM-based Evaluation Metric | | | | | | | | | |
| Qwen2.5-32B-Instruct | 59.0 | 64.7 | 59.0 | 52.3 | 55.6 | 63.0 | 64.7 | 64.7 | 65.7 |
| Qwen2.5-72B-Instruct | 59.8 | 65.7 | 60.2 | 52.2 | 57.1 | 61.9 | 61.6 | 67.5 | 63.4 |
| Llama-3.1-70B-Instruct | 59.7 | 61.4 | 53.2 | 51.0 | 55.0 | 61.2 | 60.5 | 62.6 | 58.0 |
| GPT-4o | 60.5 | 65.1 | 60.4 | 55.9 | 58.1 | 63.0 | 64.4 | 64.4 | 65.5 |
| Deepseek-V3 | 52.6 | 64.9 | 52.8 | 51.9 | 54.4 | 62.7 | 64.4 | 64.7 | 57.1 |
| RM-based Evaluation Metric | | | | | | | | | |
| Skywork-Reward-Llama | 55.9 | 57.1 | 54.1 | 55.9 | 56.2 | 57.1 | 58.0 | 63.1 | 53.5 |
| Skywork-Reward-Gemma | 56.9 | 59.0 | 59.2 | 56.3 | 56.8 | 57.0 | 58.5 | 56.6 | 60.0 |
| RM-R1-Qwen2.5-Instruct-14B | 56.9 | 59.0 | 59.2 | 56.3 | 56.8 | 57.0 | 58.5 | 56.6 | 60.0 |
| RM-R1-DeepSeek-Distilled-Qwen-14B | 65.7 | **72.1** | 64.5 | 63.9 | 61.3 | **67.7** | 73.0 | 71.4 | 69.0 |
| LRM-based Evaluation Metric | | | | | | | | | |
| o1-mini | 56.2 | 64.7 | 54.0 | 54.0 | 58.7 | 64.3 | 67.3 | 62.6 | 57.0 |
| Deepseek-r1 | 54.0 | 58.6 | 54.3 | 48.3 | 47.1 | 58.8 | 56.3 | 62.6 | 56.1 |
| Trained Evaluation Metric | | | | | | | | | |
| Auto-J-6B-bilingual | **72.3** | 68.3 | **73.1** | **65.2** | 64.3 | 67.1 | **67.4** | 69.6 | **72.9** |
| Promethus-7B-v2.0 | 66.8 | 66.4 | 69.0 | 65.1 | **65.1** | 63.9 | 64.0 | **72.0** | 57.0 |
| M-Promethus-14B | 64.1 | 63.5 | 62.0 | 59.5 | 55.2 | 62.1 | 64.9 | 70.6 | 65.5 |

framework is the most effective method for LFQA evaluation, and majority voting also contributes to improvement.

Table 13: More baselines on LFQA-E.

| Model | LFQA-E-EN | | LFQA-E-ZH | | Average | |
|---|---|---|---|---|---|---|
| | Acc | F1 | Acc | F1 | Acc | F1 |
| *SOTA Models* | | | | | | |
| Gamini2.5-flash | 63.3 | 45.6 | 57.9 | 44.7 | 60.6 | 45.2 |
| GPT-5 | 64.8 | 47.7 | 60.1 | 46.7 | 62.5 | 47.2 |
| *FactScore variants* | | | | | | |
| FineSurE (GPT-4o) | 47.8 | 35.5 | 42.9 | 32.5 | 45.4 | 34.0 |
| *Evaluation Ensembles* | | | | | | |
| ChatEval (GPT-4o) | 70.7 | 48.0 | 49.9 | 40.3 | 60.3 | 44.2 |
| G-Eval (GPT-4o) | 68.0 | 50.0 | 55.2 | 42.2 | 61.6 | 46.1 |

Table 14: Performance of different evaluation metrics on LFQA-E. The examples whose labels are tie are discarded for fairer comparison. The largest value is denoted using **bold**.

| MODEL | LFQA-E-EN | LFQA-E-ZH | Avg |
|---|---|---|---|
| Static Evaluation Metric | | | |
| Length | 42.7 | 56.9 | 49.8 ↑ 2.1% |
| ROUGE | 57.5 | 53.8 | 55.7 ↑ 3.1% |
| BERTScore | 56.0 | 56.6 | 56.3 ↑ 3.0% |
| LLM-based Evaluation Metric | | | |
| Qwen2.5-32B-Instruct | 66.8 | 62.8 | 64.8 ↑ 4.7% |
| Qwen2.5-72B-Instruct | 63.4 | 57.6 | 60.5 ↑ 3.4% |
| Llama-3.1-70B-Instruct | 66.1 | 34.0 | 50.1 ↑ 4.9% |
| GPT-4o | 66.3 | 61.4 | 63.9 ↑ **6.4**% |
| Deepseek-V3 | 60.0 | 60.0 | 60.0 ↑ 4.1% |
| RM-based Evaluation Metric | | | |
| Skywork-Reward-Llama | 56.7 | 58.4 | 57.6 ↑ 3.6% |
| Skywork-Reward-Gemma | 58.2 | 52.5 | 55.4 ↑ 3.2% |
| RM-R1-Qwen2.5-Instruct-14B | 67.5 | 56.2 | 61.9 ↑ 3.5% |
| RM-R1-DeepSeek-Distilled-Qwen-14B | 68.0 | 57.5 | 62.8 ↑ 3.3% |
| LRM-based Evaluation Metric | | | |
| o1-mini | 67.3 | **64.2** | **65.8** ↑ 4.9% |
| Deepseek-R1 | 61.6 | 63.1 | 62.4 ↑ 3.7% |
| Trained Evaluation Metric | | | |
| Auto-J-6B-bilingual | **70.0** | 57.3 | 63.7 ↑ 4.3% |
| Prometheus-7B-v2.0 | 66.5 | 54.1 | 60.3 ↑ 3.1% |
| M-Prometheus-14B | 63.7 | 53.3 | 58.5 ↑ 3.4% |

## B.5 LLMs ARE BETTER AT FINDING SOMETHING BETTER.

Considering that giving a tie option is difficult for both humans and models, we drop out the records that are labeled as a tie and conduct the experiments again. We show the results in Table 14. After discarding the tied comparison, all the evaluation metrics show nontrivial performance boosts. GPT-4o even gets a 6.4% bonus. This increase matches what we find when comparing indicators. Similar to what we observe above, LRMs remain leading on the fairer comparison, and RMs still struggle to generalize to long-form response evaluation. Static evaluation metrics, however, show the least improvement. The experimental results demonstrate the potential of test-time scaling, while reflects the generalization problem of RMs. What's more, specific evaluation models show their great potential once again, ranking first on LFQA-E-EN, displaying its future for LFQA evaluation.

## B.6 How to balance the cost and time efficiency between models?

To further discuss the tradeoff between efficiency and effectiveness among language models, we calculate the average inference time per question of several LLMs, LRMs and RMs. Table 17 lists the results. Specifically, for the closed-source models that need to invoke api for inference, we calculate their costs on both LFQA-E-EN and LFQA-E-ZH in Table 18. We observe that **o1-mini** incurs the highest cost, which is aligned with its relatively higher accuracy and F1 scores. **DeepSeek-V3**, on the other hand, offers a more cost-effective alternative, though its performance is somewhat lower. For a more practical and cost-efficient evaluation, models like Skywork-Reward-Llama and Skywork-Reward-Gemma have much smaller parameter scales and can be deployed on local resources, showing strong promise for reducing costs while still providing a reasonable evaluation.

## C DISCUSSION

We suggest training open-domain RMs or evaluation models, which may help for the evaluation of LFQA, considering their relatively low cost and GPU requirements, but with a decent score. Also, we recommend future work to focus on evaluation workflows that combine the strengths of both LLM-based models and more efficient reward models. Particularly, for open-sourced models, we don't observe steady performance gains as model size scales. Therefore, smaller models with more training data may help more than larger models with some well-designed prompts.

## D CASE STUDY

### D.1 CASE FROM CEESQ AND PEEQ

We provide two cases from CEESQ and PEEQ in Table 15. We translate them into English for easier comprehension.

Table 15: Case Study from CEESQ and PEEQ.

| |
|---|
| **QUESTION FROM PEEQ:**
What are the classifications and percentages of white blood cells? |
| **REFERENCE:**
White blood cells are divided into granular cells and non-granular cells. Granular cells include: neutrophils (50%-70%); basophils (0-1%); eosinophils (0.5%-5%). Non-granular cells include: monocytes (3%-8%); lymphocytes (20%-40%) |
| **QUESTION FROM CEESQ:**
Analyze the natural reasons for the numerous sandbars in the Yangtze River estuary area. |
| **REFERENCE**
The river has a large discharge volume and carries a large amount of sediment; located at the river estuary, the terrain is low and flat, the flow velocity is slow, with deposition as the main process, leading to massive sediment accumulation; situated at the river-sea interface, tidal backing enhances the deposition process, forming numerous sandbars in the estuary area. |

### D.2 CASE FOR PARAPHRASING

We provide a case study from LFQA-E-ZH for paraphrasing in Table 16.

Table 16: Case Study for parahrasing.

| |
|---|
| **QUESTION FROM LFQA-E-ZH:**
Reasons for the extremely fragile ecological environment in Guizhou Province. |
| **ORIGINAL RESPONSE:**
1. Guizhou features karst topography with undulating terrain
2. Serious soil erosion
3. Guizhou Province has few plains and is susceptible to natural disasters
4. Rugged terrain with inconvenient transportation |
| **PARAPHRASED RESPONSE:**
The ecological environment of Guizhou Province is extremely fragile for four main reasons: First, Guizhou features karst topography with undulating terrain and complex landforms; Second, serious soil erosion has caused severe depletion of land resources; Third, Guizhou Province has few plains, with most areas being mountainous, making it susceptible to natural disasters; Fourth, the rugged terrain with inconvenient transportation creates certain difficulties for ecological environment protection and restoration. |

Table 17: The average inference time of different language models per question, arranged in descending order.

| Model | TIME (SECONDS) |
|---|---|
| Skywork-Reward-Llama | 0.21 |
| Qwen2.5-32B-Instruct | 0.27 |
| Skywork-Reward-Gemma | 0.64 |
| Qwen2.5-72B-Instruct | 1.03 |
| o1-mini | 1.80 |
| GPT-4o | 3.56 |
| DeepSeek-V3 | 12.00 |
| DeepSeek-R1 | 14.13 |

Table 18: Overall cost on closed-source models using API. The largest value is denoted in **bold**.

| Model | LFQA-E-EN | LFQA-E-ZH |
|---|---|---|
| DeepSeek-V3 | 3.6$ | 0.7$ |
| DeepSeek-R1 | 18.0$ | 6.9$ |
| o1-mini | **67.0**$ | **14.2**$ |
| GPT-4o | 35.6$ | 5.7$ |

### D.3 FAILED CASE

We show error cases in Table 19, Table 20, and Table 21. Table 19 shows the *Incorrect Information Error*. Table 20 shows the *Point Identification Error*. Table 21 shows the *Format Error*.

### D.4 SUCCESS CASE

We show successful cases in Table 22 and Table 23.

## E INSTRUCTIONS

The following are instructions we used.

### E.1 DATA COLLECTION INSTRUCTIONS

The instruction for the question filter is in Table 24. The instruction for the paraphrase is in Table 25.

### E.2 ENGLISH LLM EVALUATION INSTRUCTION

The instruction for all LLMs and LRMs is in Table 26. The instruction for Prometheus series is in Table 27. The instruction for Auto-J is in Table 28.

## F ANNOTATION

We show the annotation recipe in Table 29 and Table 30. We show a screenshot of annotation pipeline in Figure 5.

Table 19: Case Study from LFQA-E from GPT-4o. This results from an incorrect information error.

---

**QUESTION:**

How does law enforcement track down a criminal during a manhunt?

---

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

---

**RESPONSE 1:**

There are a lot of ways that law enforcement can track someone during a manhunt. First and foremost, we live in the age of technology, and technology is a really powerful tool for tracking someone. A cell phone is pretty much always transmitting your approximate location, and law enforcement can gain access to the satellite that keeps track of that location.

There are also a few tried and true methods: the public. Law enforcement will almost always provide a physical description of a suspect, complete with a photograph or artist rendition, and urge people to report anyone who fits the bill close enough to be the person. This is a powerful tool when to get anything modern, you need to interact with other humans or come into close contact with human society.

There are also recordings, photographs, and footage captured by CCTV cameras everywhere. Police may also deploy tracking hounds if they know who you are and have anything that has your smell on it and escaping on foot from persistent, trained hunting dogs is unbelievably difficult."

---

**RESPONSE 2:**

Ohhhhh so many ways. Really depends on what resources are on hand.

1. Suspect probably has friends and family. Check their houses, interview them, see what they know about the suspects hobbies and motives. Do they have a goal they're trying to achieve? Is that goal local or would it send them running elsewhere?

2. Flowing from above, roadblocks. If you generally know where your suspect is, where they might try to go, and what means of transportation are available to them, you can monitor those points of access until they show up.

3. Technology. Where did the suspect use their credit card. Oh, they pulled out cash to avoid being traced? We can see where they made the withdrawal and go from there. Is their phone still online? Can we call and trace them? Were they recently active on social media? How about their friends? A lot of those posts have geotags in the metadata

---

**MODEL JUDGE:**

- Response 1 aligns well with the standard answer by emphasizing technology (cell phone tracking), public involvement (providing descriptions and photos), and surveillance methods (CCTV cameras and tracking dogs). It touches upon finding the suspect's connections, similar to the standard answer's first point about identifying likely places to "go to ground."

- Response 2 also covers relevant points such as investigating friends and family, setting up roadblocks, and using technology like phone tracing and social media analysis. However, it delves into areas like credit card tracking, which the standard answer specifically mentions as ineffective.

Overall, **Response 1** more closely mirrors the strategies outlined in the standard answer. (*The Judgement is **incorrect** by incorrectly estimating the weights of irrelevant or unimportant keypoints.*)

---

Table 20: Case Study from LFQA-E from GPT-4o. This results from an identification error.

**QUESTION:**
Why do swear words exist? What is it about swear words that make us feel better? Why don't other normal words make us feel the same way swear words do?

**REFERENCE:**
Swear words fall broadly into two categories: profanity and obscenity.Profanity originates with the Ten Commandments, specifically "Do not take the Lord's name in vain" in Abrahamic religions. Generally the idea though is "don't disparage or devalue the sacred". This is where the term "curse words" comes from, as you would curse someone by saying "God damn you". It's also where the term "swear words" comes from, i.e. "I swear to God". Obscenity, by contrast, is primarily words that relate to sex or using the toilet. These are activities during which you would be vulnerable to being attacked, and also that produce excrement that could be harmful for others. There is sometimes a need to discuss these things in polite company, so terms were divided into "polite" and "rude". In English, this distinction largely comes from the language of origin. English has linguistic roots in both Romance languages (ones based on Latin like French) and Germanic languages. At the time when the language was developing, higher society had more background in Romance languages while the lower classes were more Germanic. So the former words were considered refined and the latter crude and crass. Most English obscenities have close analogs in German. In modern usage, there's actually official definitions of profanity and obscenity by the FCC. Profanity is using these words in an exclamatory sense, while obscenity is using these words to describe the actions. The former is more allowable than the latter. As for why it feels good to swear, it's mostly because breaking a taboo gives a feeling of exhilaration, and if nobody actually suffers as a result, you also get to feel like you're "getting away with it". Basically, if we didn't put so much emphasis on NOT saying the words, it wouldn't feel so good when we do say them.

**RESPONSE 1:**
Language is a tool of communication. Expletives communicate something that you basically cannot express using polite language. There's certainly an element to them where the social transgression is part of what makes them function as they do. What is or isn't considered a swear word largely depends on cultural values. In the Middle Ages it's pretty apparent that words related to bodily functions or sexuality were not considered particularly offensive, while blasphemies were generally considered a lot more offensive than they are in most modern societies. Medieval folks appear to have been simultaneously less prudish but more fearful of the almighty.

**RESPONSE 2:**
Language has been used to separate classes for a very long time. Do you know why a pig is a pig in the pen but pork on your plate? Or why a cow becomes beef when you serve it up? Because one word was used by the lower class who raised the animals and the other word was used by the upper class who could afford to consume it (in this case, Normans). Words used by the lower classes frequently, but not by upper classes, will be found to be taboo in the upper classes, leading them to be essentially banned in polite society. They have power because they are taboo, and for no other reason. The word 'merde' has no power to you unless you speak French, and the word 'crap' has no power to you unless you're 6. You've been conditioned to believe these words are strong, so you feel strong when saying them. It really is that simple. It's nothing more than the placebo effect.

**MODEL JUDGE:**
Upon evaluating both responses against the reference answer:
- Response 1 aligns closely with the standard answer by discussing the role of swear words in communication, the significance of social transgression, and cultural variations in what is considered offensive. It also touches on historical perspectives, similar to how the reference answer differentiates profanity and obscenity based on origins and societal values.
- Response 2 introduces the concept of language separating classes and the power of taboo stemming from class distinctions. While it addresses the conditioning and perceived strength of swear words, it deviates more from the primary focus of the standard answer, which centers on cultural and linguistic origins rather than class-based language differentiation.
Therefore, **Response 1** is better.(*The Judgement is **incorrect** by observing the number of information coexisting.*)

Table 21: Case Study For LFQA-E from GPT-4o. This results from a format error.

**QUESTION:**
Can someone explain Benford's Law. I get that certain numbers show up more often in large data sets, but why?

**REFERENCE:**
Benford's Law is typically stated about data that span multiple orders of magnitude; so we have values that are many times larger than others instead of them being close together. Next is that we have at least a bit of what is called scale-invariance: unlike many simpler probability problems, we don't assume that all numbers are equally likely, but instead that larger ones are rarer. More precisely, we want roughly the same chance to land between x and 2x as for landing between y and 2y. If those two are satisfied, Benford's Law states that the leading digit 1 is more likely than an other, and the larger digits are the less common ones. There can be no perfect examples of scale-invariance in probability. But between those lie dozens of orders of magnitude, easily more than enough within reason...

**RESPONSE 1:**
Benford's law says if you gather up a bunch of random numbers that appear "in the wild" and you tally up the first digits of each of these numbers then you will tend to find lots of 1's and not many 9's. This tends to happen when you have a range of numbers which is spread out over multiple orders of magnitude. The reason is when numbers are spread out over many orders of magnitude, it tends to be roughly uniform when measured on a "log scale". This means that the number of data points between x and 2x will be about the same for different values of x. In other words, there should be as many values between 100 and 200 as there are between 200 and 400.

**RESPONSE 2:**
Benford's law says the first digit of a number picked from a large range of numbers tends to be a 1. Think of it like this between 1 and 20, 11 of the 20 numbers start with a 1. Between 1 and 99, 11 of the 99 numbers start with a 1. This repeats for 0-200 and 0-999 etc always holding the max of "just over half" and the min of "about 11 percent" so if you average that for all ranges, you get that about 30 percent of numbers in an unknown range start with 1. Between 1 and 89, 1 number starts with 9, basically 1/89 percent. Going up to 99 brings us back to 11/99, but now 11 is the max and "almost 0" is the min, so again average it over all ranges, and you get more like 4 percent of lead numbers. You can then apply this to some fraud cases. If the numbers span multiple orders of magnitude AND should be roughly random, AND there are a lot of them, you should expect them to match Benford's law pretty well. If they don't, one of the 3 requirements is probably false. If you know the first and last are true, you can say "these probably arent actually random".

**MODEL JUDGE:**
- Strengths of Response 1:
Correctly identifies that Benford's Law applies to numbers spanning large ranges.
Explains the higher probability of 1 as a leading digit by comparing intervals which aligns with the scale-invariance idea.
Mentions the decreasing probability for higher digits.
Correctly notes the conditions for applying Benford's Law and its use in fraud detection.
- Weaknesses of Response 1:
Doesn't explicitly mention scale-invariance, though the intuition is somewhat captured.
The explanation for why 1 is more likely is a bit simplistic.
- Strengths of Response 2: Introduces Benford's Law in a simple, relatable context. Correctly notes that 1 and 2 are more common as leading digits.
- Weaknesses of Response 2:
Misrepresents the core reasons for Benford's Law. The explanations given are incorrect or misleading. Benford's Law is not about human behavior but about inherent properties of scale-invariant data.
Doesn't mention orders of magnitude or scale-invariance at all, which are central to the reference answer.
The references at the end are irrelevant to the actual content of the response.
*No preference provided!*

Table 22: Case Study For LFQA-E from Deepseek-V3.

**QUESTION:**

how do metals put together with electricity cause technology and chips and computers and data etc?

**REFERENCE:**

Silicon along with a little bit of some other elements combined form what is called a semiconductor, of which there are 2 types, the n type and the p type. When you then sandwich these semiconductors in a specific way, it has the property that applying a current to these sandwiches causes it to become conductive, or if you invert the sandwich it has the property that it's conductive when you don't apply a current and then not conductive when you do. So essentially, what you have created is an electrical switch, which is controlled by electricity, and we call these transistors.

If you're looking at how these transistors then form a computer, you can arrange transistors to perform logical functions. A simple one would be the NOT gate, which just uses the second switch I mentioned that turns off when there is an input. An AND gate would be if you took two switches, placed them one after the other, and each switch is then controlled by an input. Both inputs need to be on for there to be an output.

**RESPONSE 1:**

For that, you need to look at boolian values and at turing machines. A boolian value is just a yes no answer. Yes vs no, on vs off, one vs zero. Technically you use boolian values all the time in speech. "Is Tommy in the bathroom?" You could answer "Yes", but you could also say "one", and anyone who knows Boolian will understand.

A Turing machine is just a name for any machine that uses Boolian values. It has rules, for example "if you see seven zeros in a row followed by a one, then do". It seems pretty abstract. But math people have actually proved that you can do any calculation, think any thought, perform any task, if you have the right rules designed for your turing machine.

So once we figured that out, it just became a matter of inventing the right rules for the machine, and inventing the best form for a machine to store boolian values. A transistor is always either a One or a Zero, and they can be made very small, so it makes sense to use them for your Turing Machine.

**RESPONSE 2:**

When metals are put together with electricity in the context of technology, it typically refers to the process of creating integrated circuits or microchips. These chips are made up of layers of different metals, such as silicon, aluminum, and copper, that are carefully arranged on a silicon wafer.

Electricity is used to power the various components on the chip, allowing them to perform their specific functions. The flow of electricity through these metal layers enables the chip to process and store information, ultimately allowing computers and other electronic devices to function.

Additionally, metals such as gold and copper are used in the construction of circuit boards and connectors within electronic devices. These metals have excellent conductivity properties, which allow for the efficient flow of electricity throughout the device.

**MODEL JUDGE:**

Response 1 delves into the concept of Boolean values and Turing machines, which extends the topic to how computers "think" and process data. It links the use of transistors to Turing machines and emphasizes the binary nature of data processing, which aligns well with the reference answer's explanation of logical functions and binary systems accessible through transistors.

Response 2 focuses on the physical aspects of creating chips, such as the use of various metals and their conductivity. While it provides a general overview of chip construction and functionality, it lacks depth on the conceptual workings, like the function of transistors and binary systems, which is a significant aspect of the reference answer.

The reference answer provides an explanation centered around the role of transistors as switches and their organization to perform logical operations. It links this to the representation of numbers and data processing in computers through binary systems, which Response 1 addresses more comprehensively compared to Response 2.

Therefore, **Response 1** is better. *(The evaluation is correct!)*

Table 23: Case Study For LFQA-E from Deepseek-V3.

**QUESTION:**
How did human calculate time?

**REFERENCE:**
For a long time we didn't and no one cared. However, once you start farming, you need to know what season it is, what hour of the day it is more or less, so you know what you can and should do on the fields for a bountiful yield. The easiest way to do is to watch the celestial bodies, namely the two major ones, the Sun and the Moon (the Stars also helped, but eventually it was more of a tool for navigation). Over time people figured out that the Sun shifts, days are not evenly distributed over the year, but luckily it goes in cycles, so it is predictable. Once you make a chart of these shifts throughout the year, you got yourself more or less a calendar. Civilizations with accurate-enough calendars survived and thrived. Calendars got more and more accurate over time as our math and science got better and here we are, keeping time down to split seconds. By the way, it does not fit perfectly. We accumulate quite the error over time, but the Gregorian Calendar is designed as a way to correct for these accumulative errors. The Persian Solar Calendar is the most accurate one, because the new year starts after a celestial event. Gregorian calendar has a bigger focus on religious events rather than any celestial event, so it needs corrections. We have a very accurate time measurement, but that is NOT what is used on the daily, because we don't need it.

**RESPONSE 1:**
Absolutely nothing about the way humans decided to calculate time "fits neatly" into Earth's behaviours. Years are actually 365 days plus 6 hours and 9 minutes, days are actually 23 hours and 56 minutes long.Ot her time systems have been attempted, with the Romans dividing day and night into 12 hours, which varied in actual length as the days and nights shifted through the year. They also had the "Civil Day" system: formal names given to time according to certain markers (such as separate named times for "rooster is crowing" and "rooster stops crowing", or "time we light candles" and "bed time"). Humans have been struggling with measuring it forever.

**RESPONSE 2:**
Time has been calculated and measured in various ways throughout history. One of the earliest methods used by ancient civilizations was observing the movement of celestial bodies such as the sun, moon, and stars. By tracking the positions of these objects in the sky, early astronomers were able to develop calendars and timekeeping systems. The modern concept of time measurement is based on the International System of Units (SI). In terms of dividing the day into hours, minutes, and seconds, this system was developed by the ancient Babylonians and Egyptians. The Babylonians used a base-60 system for counting, which led to the division of the hour into 60 minutes and the minute into 60 seconds. The Egyptians also used a similar system of timekeeping. Overall, the calculation of time has evolved over centuries through a combination of astronomical observations, scientific advancements, and cultural practices. The precise measurement of time today is achieved through the use of atomic clocks and coordinated universal time (UTC) to ensure accurate timekeeping worldwide.

**MODEL JUDGE:**
The reference answer emphasizes the historical development of time measurement, focusing on the transition from no precise timekeeping to the necessity of using celestial bodies for farming. Response 1 discusses the inconsistency and challenges of time measurements historically, noting the Roman system and difficulties faced by humans in tracking time over time. While it touches on early attempts to measure time, it doesn't delve specifically into celestial observations or the evolution of calendars as much.
Response 2 covers the historical aspect of using celestial bodies and tracks the progression to modern timekeeping with atomic clocks. It also discusses the Babylonian and Egyptian contributions to dividing time. This response aligns more closely with the standard answer by tracing the transition from ancient methods to modern precision, even mentioning cultural influences on timekeeping.
Overall, while both responses provide historical context, Response 2 is closer to the reference answer as it better follows the progression of timekeeping from ancient observations leading to the precise systems we have today.
Therefore, **Response 2** is better. *(The evaluation is correct!)*

Table 24: Prompt used for LLM-based question filtering.

---

PROMPT FOR LLM FILTER

---

**Question Filtering Instructions**

**Objective**
Filter out questions that are either unclear in their description or too broad to provide a meaningful reference.
**Filtering Criteria**

**1. Unclear Questions**
Reject questions that exhibit:
    * Ambiguous wording or phrasing
    * Multiple possible interpretations
    * Missing critical context or parameters
    * Vague or undefined terms
    * Grammatical issues that obscure meaning
    * Incomplete or fragmented thoughts

**2. Overly Broad Questions**
Reject questions that:
    * Request information on topics with no reasonable boundaries
    * Would require encyclopedic or book-length answers
    * Ask for opinions on vast, multi-faceted subjects
    * Lack specific focus or scope constraints
    * Would yield references too general to be useful
    * Cover multiple unrelated topics simultaneously

**Process**
1. Read the question carefully and completely
2. Evaluate against both clarity and breadth criteria
3. Make a filtering decision:
    **PASS**: Question is clear and appropriately scoped
    **REJECT - UNCLEAR**: Question lacks clarity (provide specific reason)
    **REJECT - TOO BROAD**: Question is overly broad (provide specific reason)

**Examples of Questions to Reject**
    * "What about technology?" (unclear)
    * "Explain everything about human history" (too broad)
    * "How does stuff work in general?" (both unclear and too broad)
    * "What are all the factors affecting everything in the world?" (too broad)

**Examples of Questions to Pass**
    * "What is the boiling point of water at sea level?"
    * "How does photosynthesis work in green plants?"
    * "What were the main causes of World War I?"

---

Table 25: Prompt for LLM Paraphrase.

---

**PROMPT FOR LLM PARAPHRASE**

---

**Objective:**
Transform the provided response into a more verbose version while strictly preserving the original meaning and information.
**Requirements:**
- Expand the original text by adding descriptive language, elaborations, and explanatory phrases
- Maintain complete fidelity to the original information—do not introduce any new facts, claims, or insights
- Preserve the tone and intent of the original message
- Use stylistic techniques such as:
   * Adding clarifying phrases and parenthetical explanations
   * Employing more elaborate sentence structures
   * Incorporating synonyms and varied vocabulary
   * Adding transitional phrases between ideas
   * Expanding brief points into full explanations
- Ensure the final text feels natural and not artificially inflated

**Process:**
1. Thoroughly analyze the original response to understand its complete meaning
2. Identify core points and supporting details
3. Expand each point methodically while maintaining the original structure
4. Review to confirm no new information has been introduced
5. Polish the text for readability and flow

---

Table 26: Prompt for English LLM Evaluation.

---

**PROMPT FOR ENGLISH LLM EVALUATION**

---

We have the following question:
Question: question
The reference (standard) answer to this question is as follows: reference
We now have two student responses to this question.
Response 1 is as follows: resp1
Response 2 is as follows: resp2
Now, you should evaluate the two responses based on the content of the question, using the standard answer as the sole basis for judgment.
Determine which of the two—Response 1 or Response 2—is closer to the reference answer in terms of content.
Please begin with a brief analysis, and then provide your final judgment in one of the following forms:
If one response is better:
"Therefore, [Response 1] is better." or "Therefore, [Response 2] is better."
If the two responses are roughly equal in quality:
"Therefore, [Both responses are equal]."

---

Table 27: Prompt for Prometheus Evaluation.

---

**PROMPT FOR PROMETHEUS EVALUATION**

---

**Task Description:**
An instruction (might include an Input inside it), a response to evaluate, and a score rubric representing a evaluation criteria are given.
1. Write a detailed feedback that assess the quality of two responses strictly based on the given score rubric, not evaluating in general.
2. After writing a feedback, choose a better response between Response A and Response B. You should refer to the score rubric.
3. The output format should look as follows: "Feedback: (write a feedback for criteria) [RESULT] (A or B)"
4. Please do not generate any other opening, closing, and explanations.
**Instruction:** {orig_instruction}
**Response A:** {response_A}
**Response B:** {response_B}
**Score Rubric:** {score_rubric}
**Feedback:**

---

Table 28: Prompt for Auto-J Evaluation.

---

**PROMPT FOR AUTO-J EVALUATION**

---

You are a helpful and precise assistant for checking the quality of the feedback.

Two pieces of feedback have been provided for the same response to a particular query. Which one is better with regard to their correctness, comprehensiveness, and specificity to the query?
**[BEGIN DATA]**
[Query]: {prompt}
[Response]: {response}
[Feedback 1]: {feedback1}
[Feedback 2]: {feedback2}
**[END DATA]**

Please choose from the following options, and give out your reason in the next line.
A: Feedback 1 is significantly better.
B: Feedback 2 is significantly better.
C: Neither is significantly better.

---

Table 29: Annotation recipe for deciding valid references of LFQA-E.

| ANNOTATION RECIPE FOR DROPPING INVALID REFERENCES |
|---|

**Goal**

Keep only real, helpful explanation. Discard those that are uninformative, incorrect, or not actual explanations.

**Keep if the answer:**
- Directly answers the question.
- Gives a simple but meaningful explanation (even if simplified).
- Is factually reasonable — not misleading or false.
- Stands alone (no "see link" or "I don't know").

**Good example:** "We yawn to help cool the brain and stay alert. It brings in oxygen and improves blood flow."

**Discard if the answer is:**
- **Not an explanation** – e.g., "Google it," "It's magic," jokes, memes, one-liners ("Because science").
- **Irrelevant** – Doesn't address the question or misunderstands it.
- **Factually wrong** – Clear misinformation (e.g., "Rain comes from clouds crying").
- **Too vague** – No real content: "It's complicated," "There are many reasons."
- **Avoids answering** – "Great question!", "Not sure, but here's a thought..." with nothing useful.
- **Circular** – Repeats the question: "We sleep because we're tired."
- **Inappropriate** – Offensive, harmful, or unprofessional content.

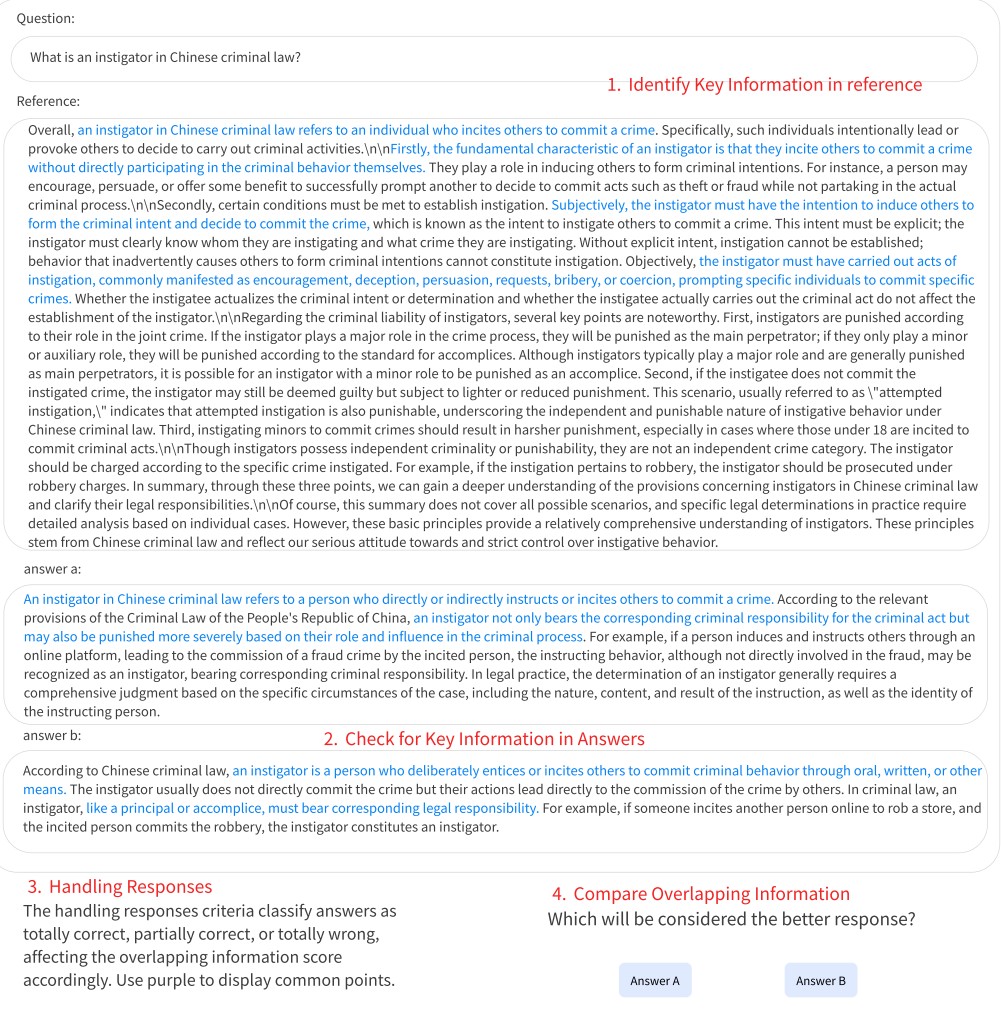

Figure 5: The annotation pipeline.

Table 30: Annotation recipe of LFQA-E.

**Overview**
This guide helps annotators evaluate and compare long-form responses against a reference to determine which response is more informative and complete. The process uses a triple-choice format (Response A Better, Response B Better, or Tie).

**Key Principles**
- Focus on **factuality** and **completeness** according to the reference
- Fluency is not a primary evaluation criterion (all responses are expected to be fluent)
- Use information units as the basic evaluation unit
- Minimize bias through systematic comparison

**Prerequisites**
- Domain knowledge relevant to the question topic
- Understanding of the subject matter through academic coursework or professional experience
- Ability to maintain focus during paragraph-level analysis

**Evaluation Process**

**Step 1: Extract Key Information from Reference**
1. Read the question carefully to understand what information is being requested
2. Read the reference thoroughly
3. Identify and list all key information units that:
   - Directly answer the question
   - Provide necessary context or background
   - Support the main answer with evidence or examples
4. Organize key information into logical categories or themes

**Step 2: Check for Key Information in Responses**
For each response (A and B):
1. Read the response completely
2. Map each key information unit from the reference to the response
3. Mark which key information units are:
   - Present and accurate
   - Present but inaccurate
   - Missing entirely
4. Note any additional information not in the reference

**Step 3: Handle Response Content**
1. Evaluate additional information:
   - Is it relevant to the central topic?
   - Does it enhance understanding or is it verbose/unnecessary?
2. Identify intertwined information:
   - For sentences containing both correct and incorrect information, separate the components
   - Assess the impact of any inaccuracies on the overall response quality

**Step 4: Compare Overlapping Information**
1. Compare how well each response covers the key information units
2. Consider:
   - Completeness: Which response includes more key information?
   - Accuracy: Which response presents information more correctly?
   - Relevance: Which response stays more focused on the question?
3. Compare the quality of overlapping information presentation

**Step 5: Make Final Decision**
Select one of three options:
- **Response A is Better**: A contains more key information and/or presents it more accurately
- **Response B is Better**: B contains more key information and/or presents it more accurately
- **Tie**: Both responses are comparable in information coverage and accuracy

**Common Pitfalls to Avoid**
1. Losing focus due to long paragraphs - use the systematic approach
2. Allowing domain bias to influence decisions - stick to the reference
3. Confusing eloquence with accuracy
4. Missing subtle differences between comparable responses

