# OpenReview forum: "LFQA-E: Carefully Benchmarking Long-form QA Evaluation"
_ICLR.cc/2026/Conference — ICLR 2026 Poster_

### Official Review · Reviewer_aZgf · 2025-10-31

**Soundness:** 3
**Presentation:** 3
**Contribution:** 2
**Rating:** 2
**Confidence:** 4

**Summary:**

This paper presents an evaluation dataset consisting of -- 1k data with question, pairwise answer, a reference answer and a human annotated preference covering English and Chinese. The paper then conducted evaluation of different evaluation metrics' correlation with the human annotation, concluding that none of them reliably model the human preference.

Overall the paper is written in a clear manner. However I do not find it to bring substantial contribution compared to prior work in the space of evaluating evaluation metrics. Please see discussion below.

**Strengths:**

* The paper is written in a clear manner.
* The paper conducted comprehensive experiments with different types of metrics (reference-based, reward models, etc.)
* The annotation process is conducted clearly.

**Weaknesses:**

* Limited contribution: the paper aims to contribute a dataset with reference-based evaluation for long-form answers, yet there are existing resources such as Feedback-Bench from [Prometheus-2(EMNLP 2024)](https://arxiv.org/abs/2405.01535). It would be good to further clarify the new contribution this new dataset introduces. Also while the benefit of having a reference answer is understandable, it is also a somewhat unrealistic setting -- many user queries would not contain a reference answer. Also it is possible that a single reference answer does not comprehensively cover all the possible answers, especially for more open-ended questions studied in this paper.
* Annotation noise: while the paper reported a cohen's kappa of 0.77 between annotators, each question is only annotated by two annotators -- this raises the question of whether the dataset capture annotators' preference, or other aspects. On a related note, it would be good to identify what are the **aspects** that these preference capture.

**Questions:**

* Given that there are only two annotators, how is the gold label determined when the two annotators disagree?
* The prompt for LLM evaluation (table 22) appears to be incorrect.
* Figure 2 reports performance of different comparison setting (human v.s. human, model v.s. model) and shows that evaluation metrics perform worse on model v.s. model comparison -- what's the human performance for these settings?

---

> ### Author Response · Authors · 2025-11-19
>
> **(W1) Difference from Feedback-Bench and validation of setting**
>
> 1. While Feedback-Bench is a valuable resource, LFQA-E is designed more robustly and of high quality. Our contributions are threefold:
> - Robustness through Manual Curation: Unlike Feedback-Bench, which is constructed autonomously, LFQA-E is primarily built through expert manual annotation. This human-in-the-loop process ensures higher quality, nuance, and reliability in the preference labels.
> - Increased Challenge and Discriminative Power: LFQA-E is intentionally enriched with difficult comparisons. As our results show below, this makes our benchmark significantly more challenging. Leading models achieve very high scores on Feedback-Bench, but their performance drops substantially on LFQA-E, proving its ability to better differentiate model capabilities.
> - Comprehensive and Fair Design: LFQA-E is multi-lingual and includes three distinct types of comparisons to ensure a holistic and fair assessment, going beyond the single-task benchmarks.
>
> | Model | Feedback-Bench | LFQA-E |
> |-------|----------------|--------|
> | Qwen2.5-32B-Instruct | 86.8% | 60.1% |
> | Qwen2.5-72B-Instruct | 94.4% | 57.1% |
> | GPT-4o | 89.2% | 57.5% |
> | GPT-5 | 75.6% | 62.5% |
> | DeepSeek-V3 | 83.6% | 55.9% |
> | DeepSeek-R1 | 83.6% | 58.7% |
>
> 2. We agree that a reference-free setting is common, but we argue that a reference-based evaluation is essential for our specific goal: to benchmark the true discriminative capability of evaluation metrics. The reference acts as an objective "golden baseline" or rubric, grounding the evaluation and preventing metrics from rewarding plausible but incomplete or incorrect answers. This controlled setting allows us to isolate and measure the evaluation metric's performance more accurately. Furthermore, in real-world applications, modern LLMs can act as search agents to first gather relevant information, effectively creating a reference context before making a judgment.
>
> 3. We acknowledge that a single reference may not cover all valid answers for open-ended questions. But we aim to benchmark the comparison with references containing all the points needed to answer the given question. We task experts with leaving comprehensive references that cover all key points needed for a complete answer. This expert-driven curation process ensures that the reference serves as a robust and thorough standard for evaluation.
>
> **(W2) Annotation Noise**
>
> Thank you for raising this concern. While each individual question is initially annotated by two annotators, the entire dataset is labeled by a pool of 10 annotators, and for each domain, the data is annotated by annotators with professional annotation experience. When the two initial annotators disagree, a third annotator adjudicates. The reported Cohen's $κ$ of 0.77 is computed over these initial pairs within the annotator pool, thus reflecting the overall consistency of preferences rather than the idiosyncrasies of a single pair.
>
> We also perform a small post-hoc analysis on a subset of the data to identify which aspects drive the preferences:
> To investigate how different factors influence annotators' decision-making during pairwise comparisons, we randomly sample 600 comparison pairs from LFQA-E-zh and LFQA-E-en. For each comparison, we count how many factors are invoked by the ten annotators when making their final judgments. Each annotator will select the top 2 factors they care about when making a decision. The overall distribution of factor usage is summarized below:
>
> | Factor | Role in preference（Voted by all annotators） |
> |--------|-----------------------------------------------|
> | Factuality | 349 |
> | Completeness | 230 |
> | Professionalism | 229 |
> | Correct use of examples | 153 |
> | Well-structured | 130 |
> | Easy to understand | 80 |
> | Grammar | 29 |
>
> This analysis indicates that our preference labels mainly capture judgments of factuality and overall information quality, rather than annotator-specific stylistic preferences.
>
> **(Q1) Conflict between annotators**
>
> Great question. If there exists a conflict between two annotators, we invite a third annotator to justify the final answer and provide the final preference.

---

> > ### Author Response · Authors · 2025-11-19
> >
> > **(Q2) Incorrect prompt**
> >
> > Sorry for the misunderstanding. We provide the prompt for LLMs to evaluate below, and we will update it in our revised manuscript.
> >
> > ```
> > We have the following question:
> >
> > Question: {question}
> > The reference (standard) answer to this question is as follows: {reference}
> >
> > We now have two student responses to this question.
> > Response 1 is as follows: {resp1}
> > Response 2 is as follows: {resp2}
> >
> > Now, you should evaluate the two responses based on the content of the question, using the standard answer as the sole basis for judgment.
> > Determine which of the two—Response 1 or Response 2—is closer to the reference answer in terms of content.
> >
> > Please begin with a brief analysis, and then provide your final judgment in one of the following forms:
> >
> > If one response is better:
> > "Therefore, [Response 1] is better." or "Therefore, [Response 2] is better."
> >
> > If the two responses are roughly equal in quality:
> > "Therefore, [Both responses are equal]."
> > ```
> >
> > **(Q3) Human baselines across settings**
> >
> > Thank you for your insightful question. We show the human baselines of the three settings below. Our analysis shows a clear divergence from automatic metrics: while metrics struggle with model-vs-model comparisons, human evaluators perform very well in this setting. Conversely, humans find human-vs-human comparisons more challenging, which we attribute to the rich semantic meaning and diverse structures used by humans. This observation further validates our core finding that LLMs do not replicate human evaluation patterns, but instead rely on a distinct set of internal criteria.
> >
> > | Setting | LFQA-E-zh | LFQA-E-en |
> > |---------|------------------|------------------|
> > | Human vs human | 77.42 | 78.12 |
> > | Human vs model | 74.19 | 81.25 |
> > | Model vs model | **80.64** | **84.73** |

---

### Official Review · Reviewer_H1wd · 2025-10-31

**Soundness:** 3
**Presentation:** 3
**Contribution:** 2
**Rating:** 4
**Confidence:** 4

**Summary:**

- Focuses on LFQA (Long-Form Question Answering)
 - Existing LFQA benchmarks are small, narrow and often lack reference answers.
 - LFQA-E is a multilingual, reference-based benchmark with 1,625 questions and 7,649 pairwise comparisons across 15 diverse topics.
 - The dataset covers sources like online queries and exam questions, supporting broad metric evaluation.
 - The study assesses 17 automatic metrics across five categories using LFQA-E.
 - Findings: no current automatic metric aligns well with human judgment; all struggle to capture dense, nuanced information in long-form responses.

**Strengths:**

The goal of effective evaluation of long responses is clearly an important for AI/NLP.

**Weaknesses:**

Assuming that table 1 refers to tokens (Table 1 are tokens, not chars?), 180-200 tokens is not much really, compared to what modern models can generate.

For your LLM-based evaluation metrics, have you tries using prompt that involves generating intermediate thoughts before conclusing a final answer?

For reasoning models, can you please see the quality metrics as a function of the thinking budget?

There have been various benchmarks for reward modeling (input Q, and two responses with preference label: A1 > A2). These benchmarks are essentially equivalent to what you're doing here. Can you distinguish yourself against these works and what gap your work is filling?

Make sure your paragraph heading have a consistent style (e.g., they all end with "." at the end).

Relatedly, this an off way to format your paragraph heading. Like it's part of the subsequent sentence??
> **For LM-based Evaluation Metric** We observe ...

**Questions:**

See the previous box.

---

> ### Author Response · Authors · 2025-11-19
>
> **(O1) Limited context length**
>
> Thank you for your insightful comment. To clarify, Table 1 refers to words, not tokens. Though the length of response is not so long compared with the max tokens of current LLMs, in our task, the model must process and compare a reference answer, two candidate responses, and the original question. The total input length is therefore substantially greater than the length of any single response. Besides, comparing with the previous benchmark, LFQA-E shows the longest response length [1,2,3].
> | Hurdles | WebGPT | Expert | LFQA-E |
> |---------|--------|--------|--------|
> | #Words Questions | 35.0 | 17.9 | 42.0 |
> | #Words Answers | 215.1 | 135.5 | 104.0 | 245.0 |
>
> Furthermore, we posit that verification is inherently more demanding than generation [4,5]. A model can generate a plausible but flawed answer with a limited understanding, but accurate verification requires a deep, holistic comprehension of all provided texts to discern subtle differences and factual inaccuracies.
>
> **(O2) Prompt for intermediate thought**
>
> Yes. Regarding your question about the use of prompts that involve generating intermediate thoughts, we indeed employ CoT reasoning in our evaluation metrics. This approach can be specifically referenced in case studies from Table 15-18 of the paper. Particularly for LRM-based metrics, these models generate even longer chains of thought, which inherently include reasoning steps.
>
> **(O3) Relation between sample budget and accuracy**
>
> Great question. We further experiment using Qwen3-32B, which is a reasoning model empowering strong capability. We repeatedly sample 16 results for a given comparison, and have the results below, where Pass@K represents at least one correct sample occurring in the k responses.
>
> | K | 1 | 2 | 3 | 4 | 5 | 6 | 7 | 8 | 9 | 10 | 11 | 12 | 13 | 14 | 15 | 16 |
> |---|---|---|---|---|---|---|---|---|---|----|----|----|----|----|----|----|
> | Pass@K | 55.0% | 75.9% | 82.5% | 87.5% | 88.3% | 88.3% | 90.8% | 92.5% | 94.7% | 95.7% | 96.5% | 97.0% | 98.0% | 98.5% | 98.8% | 99.0% |
>
> Using the formula provided by LLM Monkey [6], we could fit a function $\log c= 1.8k^{0.04}$, where c is pass@k, and R-squared is 0.86, indicating high fitness.
>
> **(O4) Difference between LFQA-E and reward model benchmarks**
>
> Thank you for this critical question. While existing RM benchmarks share a similar structure, LFQA-E is different from them in the following aspects.
> - First, our data quality is superior. LFQA-E is built primarily through expert manual annotation and is enriched with difficult comparisons between top-performing models. This makes our benchmark more robust and challenging than those relying on automated data.
> - Second, our evaluation design is more comprehensive. LFQA-E is multi-lingual, consisting of three modes, and includes a high-quality reference for each question—a key feature missing from recent benchmarks like RM-Bench [7] and Reward-Bench [8]. We believe this reference provides an objective grounding for what is otherwise a subjective preference, making the evaluation more reliable.
> - Finally, LFQA-E is demonstrably more difficult. Our results show that leading evaluation metrics achieve high scores on existing benchmarks but struggle significantly on LFQA-E. This performance gap is strong evidence that our benchmark is more rigorous and better at revealing the true limits of current models.
>
> | Model | Reward-Bench | RM-Bench | LFQA-E |
> |-------|-------------|----------|--------|
> | Skywork-Reward-Gemma-2-27B | 75.3 | 69.5 | 52.2 |
> | Skywork-Reward-Llama-3.1-8B-v0.2 | 71.8 | 72.6 | 54.0 |
> | Gemini-2.5-flash | 77.7 | N/A | 60.6 |
> | RM-R1-DeepSeek-Distilled-Qwen-14B | N/A | 71.8 | 59.5 |
> | RM-R1-Qwen-Instruct-14B | N/A | 75.6 | 58.4 |

---

> > ### Author Response · Authors · 2025-11-19
> >
> > **(O5 & O6) Heading and formatting**
> >
> > Thank you for your comment regarding the paragraph headings. We apologize for the inconsistency and will ensure all headings follow a consistent style with proper punctuation in the revised manuscript.
> >
> > **Reference**
> >
> > [1] Fangyuan Xu, Yixiao Song, Mohit Iyyer, and Eunsol Choi. 2023. A Critical Evaluation of Evaluations for Long-form Question Answering. arXiv preprint arXiv:2305.18201.
> >
> > [2] Kalpesh Krishna, Aurko Roy, and Mohit Iyyer. 2021. Hurdles to Progress in Long-form Question Answering. arXiv preprint arXiv:2103.06332.
> >
> > [3] Reiichiro Nakano, Jacob Hilton, Suchir Balaji, Jeff Wu, Long Ouyang, Christina Kim, Christopher Hesse, Shantanu Jain, Vineet Kosaraju, William Saunders, Xu Jiang, Karl Cobbe, Tyna Eloundou, Gretchen Krueger, Kevin Button, Matthew Knight, Benjamin Chess, and John Schulman. 2022. WebGPT: Browser-assisted question-answering with human feedback. arXiv preprint arXiv:2112.09332.
> >
> > [4] Lingfeng Zhou, Jialing Zhang, Jin Gao, Mohan Jiang, and Dequan Wang. 2025. PersonaEval: Are LLM Evaluators Human Enough to Judge Role-Play? arXiv preprint arXiv:2508.10014.
> >
> > [5] Thang Luong, Dawsen Hwang, Hoang H. Nguyen, Golnaz Ghiasi, Yuri Chervonyi, Insuk Seo, Junsu Kim, Garrett Bingham, Jonathan Lee, Swaroop Mishra, Alex Zhai, Clara Huiyi Hu, Henryk Michalewski, Jimin Kim, Jeonghyun Ahn, Junhwi Bae, Xingyou Song, Trieu H. Trinh, Quoc V. Le, and Junehyuk Jung. 2025. Towards Robust Mathematical Reasoning. arXiv preprint arXiv:2511.01846.
> >
> > [6] Bradley Brown, Jordan Juravsky, Ryan Ehrlich, Ronald Clark, Quoc V. Le, Christopher Ré, and Azalia Mirhoseini. 2024. Large Language Monkeys: Scaling Inference Compute with Repeated Sampling. arXiv preprint arXiv:2407.21787.
> >
> > [7] Yantao Liu, Zijun Yao, Rui Min, Yixin Cao, Lei Hou, and Juanzi Li. 2024. RM-Bench: Benchmarking Reward Models of Language Models with Subtlety and Style. arXiv preprint arXiv:2410.16184.
> >
> > [8] Saumya Malik, Valentina Pyatkin, Sander Land, Jacob Morrison, Noah A. Smith, Hannaneh Hajishirzi, and Nathan Lambert. 2025. RewardBench 2: Advancing Reward Model Evaluation. arXiv preprint arXiv:2506.01937.

---

> > > ### Comment · Reviewer_H1wd · 2025-11-24
> > >
> > > Thanks to the authors for their clarification.
> > >
> > > You have answered most of my questions.
> > >
> > > So I think overall, this work is technically sound (or at least I don't see any issues here).
> > >
> > > However, my only hesitation to support accepting this paper is:
> > >  1. Given the average output length of ~300 words, is a far cry from the typical length of the models.
> > >  2. I have no doubt that the resulting data is going to be useful for someone. It is not clear to me that it's going to be a very exciting, needle-moving release. In other words, why should be published in a top-tier venue like ICLR? Of course, meta-reviewers will make their own recommendation on this given all this discussion.

---

> ### Author Response · Authors · 2025-11-25
>
> > 1. Limited length compared with context length of LLMs
>
> We thank the reviewer for this valuable feedback. We acknowledge the concern and wish to clarify that our definition of "long" in LFQA is grounded in specific task attributes and empirical data. We posit that the primary challenge of this work is not merely maximizing generation length, but rather complex evaluation under the LFQA setting.
>
> Our position is supported by the following three points:
>
> - Task Definition & Empirical Grounding: LFQA is distinct from "report-level" generation; it is best defined as "paragraph-level" QA aimed at providing in-depth, explanatory answers. We align our scope with the ELI5 dataset, and as noted in the WebGPT analysis, the average human answer length in ELI5 is 215.1 words. This empirical evidence suggests that high-quality, informative answers do not inherently require excessive length.
>
> - Evaluation Complexity: The core technical challenge we care about is the evaluation process, not just the response length. Our benchmark requires the evaluator model to process a complex prompt containing: [Question + Reference Answer + Candidate A + Candidate B]. This results in an input length of approximately 1300 tokens (~1000 words) per instance, posing a significant challenge to the contextual understanding capabilities of current LLMs.
>
> - Reasoning Density vs. Context Length: We emphasize a distinction between our work and benchmarks like LongBench. While LongBench stresses the retrieval of needles in massive haystacks (context length), our work prioritizes reasoning density. The evaluator must compare two nuanced responses against a reference and adjudicate quality—an inherently multi-step reasoning process. Therefore, while the text length is moderate, the reasoning depth required is substantial.
>
> > 2. Significance of the work
>
> We thank the reviewer for acknowledging the utility of our data. We would like to address the concern regarding the significance of this work.
>
> We argue that this benchmark is a crucial step toward establishing stronger evaluation metrics capable of distinguishing between nuanced responses. Importantly, our contribution is not limited to LFQA; it aims to address a fundamental bottleneck in broader model training and evaluation:
>
> - The Shift to General Domains: While RL is widely successful in domains with verifiable rewards (e.g., math, code), the next frontier is applying RL to general domains without a unique label. This transition requires devising significantly stronger verifiers (reward models). Since RL relies on policy rollouts and updates, the generated outputs for a single question often become semantically very similar as the model converges. Distinguishing which response is marginally "better" among these similar rollouts is a substantial challenge. Also, how to evaluate the model performance after a few training steps also requires a fine-grained evaluation.
>
> - Our Contribution: The above is exactly the capability our benchmark evaluates. By rigorously testing a model's ability to distinguish between competitive, nuanced responses, we provide the necessary part to advance RL research in general domains. Also, references or rubrics have been proven effective in previous works [1,2], where we move a step further compared with previous reward model benchmarks by incorporating expert-written references.
>
> [1] Shuqian Sheng, Yi Xu, Luoyi Fu, Jiaxin Ding, Lei Zhou, Xinbing Wang, and Chenghu Zhou. 2024. Is Reference Necessary in the Evaluation of NLG Systems? When and Where? arXiv preprint arXiv:2403.14275.
>
> [2] Thang Luong, Dawsen Hwang, Hoang H. Nguyen, Golnaz Ghiasi, Yuri Chervonyi, Insuk Seo, Junsu Kim, Garrett Bingham, Jonathan Lee, Swaroop Mishra, Alex Zhai, Clara Huiyi Hu, Henryk Michalewski, Jimin Kim, Jeonghyun Ahn, Junhwi Bae, Xingyou Song, Trieu H. Trinh, Quoc V. Le, and Junehyuk Jung. 2025. Towards Robust Mathematical Reasoning. arXiv preprint arXiv:2511.01846.

---

### Official Review · Reviewer_Yu2f · 2025-11-01

**Soundness:** 3
**Presentation:** 3
**Contribution:** 3
**Rating:** 8
**Confidence:** 4

**Summary:**

This paper introduces a new comprehensive evaluation benchmark for evaluating long form QA systems and models. The main contributions of this paper are as follows:
1. LFQA-E a benchmark dataset for evaluation of automatic metrics for LFQA.
2. Detailed analysis of the proposed benchmark
3. Experimental evaluation on 15 evaluation metrics to show that current LFQA evaluation metrics fail to capture core information.
4. Analysis of generalisation of different metrics across languages and settings.

**Strengths:**

The main strengths of the paper are as follows:
1.  The paper identifies important limitations of the existing LFQA benchmarks like lack of authorized references and limited diversity and proposes a new large scale benchmark for better LFQA evaluation.
2. The benchmark is very diverse and spans 15 topics and domains.
3. Human experts based validation to makes the benchmark trustworthy.
4. Detailed comparison of standard evaluation metrics on a variety of reasoning models using frontier models like deepseek and qwen family of models.
5. Detailed benchmark contamination analysis to ensure that the evaluation data is not memorised during pre-training.

**Weaknesses:**

The main weaknesses of the paper are as follows:
1. The comparisons of evaluation results using previous LFQA benchmarks using frontier models are missing.
2. The paper writing and presentation can be improved.

**Questions:**

1. Is there a metrics used to assess difficulty of the questions in the benchmark?
2. Is there a plan to release a multilingual version of this dataset?
3. Reddit license has some restrictions, do the authors plan to release the benchmark and setup a leaderboard?

---

> ### Author Response · Authors · 2025-11-19
>
> **(W1) Comparison with previous benchmarks**
>
> Thank you for this important suggestion. To demonstrate the value and distinctiveness of our LFQA-E benchmark, we conduct a comparative analysis against several established baselines using frontier models. Our findings consistently show that LFQA-E presents a more significant challenge, thereby better differentiating the capabilities of state-of-the-art models.
> - We first compare LFQA-E with the expert-annotated benchmark from Xu et al. [1] (which we term "Expert") and Feedback-Bench [2]. As shown in the table below, all leading models achieved substantially lower scores on LFQA-E. This performance gap validates that LFQA-E probes deeper into models' reasoning and evaluation capabilities.
>
> | Model                     | Feedback-Bench | Expert | LFQA-E |
> |---------------------------|----------------|--------|--------|
> | Qwen2.5-32B-Instruct      | 86.8%          | 74.2%  | 60.1%  |
> | Qwen2.5-72B-Instruct      | 94.4%          | 72.7%  | 57.1%  |
> | GPT-4o                    | 89.2%          | 70.0%  | 57.5%  |
> | GPT-5                     | 75.6%          | 68.9%  | 62.5%  |
> | DeepSeek-V3               | 83.6%          | 65.8%  | 55.9%  |
> | DeepSeek-R1               | 83.6%          | 68.9%  | 58.7%  |
>
> - We then extend our comparison to specialized reward model benchmarks, including RM-Bench [3] and Reward-Bench [4]. The results, presented in the table below, reaffirm our conclusion. The consistent performance drop on LFQA-E across various reward models highlights its robustness and difficulty, even for systems specifically designed for evaluation.
>
> | Model                                   | Reward-Bench | RM-Bench | LFQA-E |
> |-----------------------------------------|--------------|----------|--------|
> | Skywork-Reward-Gemma-2-27B              | 75.3%        | 69.5%    | 52.2%  |
> | Skywork-Reward-Llama-3.1-8B-v0.2        | 71.8%        | 72.6%    | 54.0%  |
> | Gemini-2.5-flash                        | 77.7%        | N/A      | 60.6%  |
> | RM-R1-DeepSeek-Distilled-Qwen-14B      | N/A          | 71.8%    | 59.5%  |
> | RM-R1-Qwen-Instruct-14B                 | N/A          | 75.6%    | 58.4%  |
>
> **(W2) Improved presentation and writing**
>
> Thank you for your patient reading and kind advice. We will refine our writing and presentation in our revised manuscript. If something is unclear, welcome to discuss.
>
> **(Q1) Quantifying difficulty**
>
> Thank you for this insightful question. We assess the difficulty of LFQA-E not through a single quantitative metric, but through its construct validity. The difficulty is an emergent property of two core design principles:
> 1. Information Retrieval Challenge: Long-form QA evaluation requires synthesizing information scattered across long, complex documents.
> 2. Reasoning Depth: Answers demand multi-step inference and comprehensive analysis, not simple fact retrieval.
> We obtain this through selecting student responses with similar assigned scores and models with similar capabilities. Rather than assigning a numerical difficulty score, we empirically validate the benchmark's challenge by demonstrating that it creates a significant performance gap even among SOTA models.
>
> **(Q2) Plan for open source**
>
> Yes, we do plan to release a multilingual version of the dataset. And we will hold a leaderboard for community usage.
>
> **(Q3) License and plan for leaderboard**
>
> We are aware of the restrictions associated with the Reddit license. In light of this, we plan to release the LFQA-E benchmark with proper adherence to licensing requirements. Once the benchmark is publicly available, we will establish a leaderboard to track the performance of different models on LFQA-E, fostering further advancements in LFQA evaluation.
>
> **Reference**
>
> [1] Fangyuan Xu, Yixiao Song, Mohit Iyyer, and Eunsol Choi. 2023. A Critical Evaluation of Evaluations for Long-form Question Answering. arXiv preprint arXiv:2305.18201.
>
> [2] Seungone Kim, Jamin Shin, Yejin Cho, Joel Jang, Shayne Longpre, Hwaran Lee, Sangdoo Yun, Seongjin Shin, Sungdong Kim, James Thorne, and Minjoon Seo. 2024. Prometheus: Inducing Fine-grained Evaluation Capability in Language Models. arXiv preprint arXiv:2310.08491.
>
> [3] Yantao Liu, Zijun Yao, Rui Min, Yixin Cao, Lei Hou, and Juanzi Li. 2024. RM-Bench: Benchmarking Reward Models of Language Models with Subtlety and Style. arXiv preprint arXiv:2410.16184.
>
> [4] Saumya Malik, Valentina Pyatkin, Sander Land, Jacob Morrison, Noah A. Smith, Hannaneh Hajishirzi, and Nathan Lambert. 2025. RewardBench 2: Advancing Reward Model Evaluation. arXiv preprint arXiv:2506.01937.

---

### Official Review · Reviewer_awMT · 2025-11-01

**Soundness:** 3
**Presentation:** 3
**Contribution:** 3
**Rating:** 4
**Confidence:** 3

**Summary:**

This paper introduces LFQA-E, a reference-based multilingual benchmark for evaluating automatic metrics on long-form question answering (LFQA). The motivation is that existing automatic metrics fall short on LFQA, and also there are gaps in metric generalization. The benchmark contains questions and pairwise comparisons across 15 topics in English and Chinese, with expert-written references and a triple-choice annotation protocol (A / B / tie). The authors evaluate 17 evaluation methods spanning static, LLM-based, reward/model-based, LRM, and trained evaluation models.

**Strengths:**

S1) The paper addresses an important, underexplored issue: reliable automatic evaluation for paragraph-length, information-dense LFQA answers. The motivation and relation to prior small/weak benchmarks are convincing.

S2) The paper offers a new substantial and diverse dataset for LFQA evaluation, offering 1,625 Questions and 7.6k comparisons, multilinguality (EN/ZH), and topic breadth. Statistics and careful sourcing are provided.

S3) The annotation recipe is thorough. Using experts, double annotation, a triple-choice rubric, and inter-annotator agreement reporting improves trust in labels.

S4) Authors test a wide range of metrics, give per-domain breakdowns, error taxonomy, contamination checks (PPL, n-gram overlap), and also try improvement (TTRL). This breadth of evaluation is useful for diagnosing metric weaknesses.

**Weaknesses:**

O1) The English references use the “top-ranked ELI5 answer” as candidate references. Top reddit answers are not guaranteed to be expert or correct; relying on them (even after expert review) risks reference quality variance. Although the authors state expert review, more detail is needed: how often did experts modify reddit content, and can we get quantitative measures of reference quality across sources?

O2) Annotators are paid $2 per question (4–6 comparisons). This low pay risks rushed judgments despite training. The paper reports κ, but does not report annotator qualification statistics, and time per item distribution (they say ~7 minutes per comparison — is that per comparison or per question?). These details matter for external confidence.

O3) Several choices need clarification or stronger controls: prompts, temperature settings, and inclusion of references differ across evaluated models. The procedural differences can advantage/disadvantage classes of metrics. The authors report a temperature ablation, but more systematic control (same prompt templates, prompt-sensitivity analysis, ensemble-vs-scalar handling) is needed to ensure fair comparisons.

O4) Some leading recent evaluation approaches (explainable/extractive hybrid metrics, factuality-oriented measures such as FactScore variants, or evaluation ensembles) are missing or treated superficially. It’s unclear whether the 17 chosen methods fairly represent the state of the art; the paper would benefit from explicitly listing and justifying excluded but relevant baselines.

**Questions:**

Please address the weaknesses O1-O4.

---

> ### Author Response · Authors · 2025-11-19
>
> **(O1)** Thank you for your kind feedback. We will clarify below.
> - As mentioned in line 171 of the manuscript, we don't modify the references from Reddit. Instead, experts focus on filtering out irrelevant or inappropriate references during the dataset curation process. Specifically, experts review **2236** questions and corresponding answers from the ELI5 subreddit. If they find any question-answer pair to be unsuitable or of poor quality, they discard it directly. After the expert filtering process, only **1,026** question-answer pairs remain for the final dataset.
> - To ensure the robustness of our evaluation, first, we use Cohen's Kappa to measure the inter-annotator agreement among our experts. The high Kappa score confirms the consistency and reliability of the annotation process itself. To make it clear, we further conduct a detailed quality assessment, rating the references on several key indicators from 1 - 5, where 5 is the best. As shown in the table below, the results clearly demonstrate that our filtering process significantly enhances reference quality.
>
> |               | Completeness | Factuality | Professionalism | Use of examples | Easy to understand |
> |---------------|--------------|------------|-----------------|-----------------|---------------------|
> | Pre-Filter    | 2.2          | 2.8        | 1.9             | 2.0             | 1.6                 |
> | Post-Filter   | **3.1**          | **4.6**        | **3.4**             | **2.8**             | **2.5**                 |
>
> **(O2)**  Thank you for raising the important question regarding our annotation budget and process.
> - Our compensation rate is set at 2 dollars per question, a decision made after careful consideration of regional economic standards and project budget constraints. To provide context, this rate is above the local minimum hourly wage of 1.99 dollars. Also, our statistics show a range of 3 dollars to 6 dollars per hour, and our total question budget is consistent with that of Xu et al. [1]. Besides, to ensure annotation quality, we carry out a preliminary annotation experiment to ensure that each annotator is familiar with the task, thus enhancing the quality. The final substantial annotation consistency also reflects the high quality of the annotation, considering the difficulty of such a comparison.
> - To ensure quality and fairness, we provide the annotator qualification statistics below, including major, education experiences, and annotation consistency before, which indicate a rather high level.
>
> | No | Major                | Education | Previous Consistency |
> |----|----------------------|-----------|---------------------|
> | 1  | Computer science     | PhD       | 90.39               |
> | 2  | Basic Medicine       | PhD       | 89.51               |
> | 3  | Mathematics          | PhD       | 87.16               |
> | 4  | Electrical Engineering | PhD    | 88.04               |
> | 5  | Theoretical Economics | PhD     | 88.56               |
> | 6  | Law                  | MS        | 90.18               |
> | 7  | Psychology           | MS        | 85.9                |
> | 8  | Geology              | Bachelor  | 86.42               |
> | 9  | Finance              | Bachelor  | 85.4                |
> | 10 | History              | Bachelor  | 83.12               |
>
> - Yes. Each comparison takes around 7 minutes, and they share the same references.

---

> > ### Author Response · Authors · 2025-11-19
> >
> > **(O3)** Great question. To address these concerns, we have conducted additional ablation studies and wish to clarify our methodological choices point-by-point.
> > - Above all, we wish to clarify that all LLM-based and LRM-based methods are evaluated using the exact same prompt template. The only exception was for fine-tuned metrics, which, by their nature, require the specific prompt template they are trained on to achieve optimal performance.
> > - To further investigate the sensitivity to prompt design, we conduct an additional ablation by instructing models to solely provide a final answer without CoT. The results are presented in the table below. This experiment further highlights the importance of CoT for enhancing the ability when evaluating: for most settings under LFQA-E-EN, the performance degrades greatly. While for LFQA-E-ZH, the performance is hard to predict, indicating the language vulnerability.
> >
> > | Model                     | LFQA-E-EN       |                 | LFQA-E-ZH       |                 |
> > |---------------------------|-----------------|-----------------|-----------------|-----------------|
> > |                           | Acc             | F1              | Acc             | F1              |
> > | Qwen2.5-32B-Instruct      | $57.3_{↓6.2}$      | $45.5_{↓0.3}$       | $52.8_{↓3.9}$      | $38.6_{↓3.2}$       |
> > | Qwen2.5-72B-Instruct      | $60.4_{↓0.8}$      | $45.1_{↑2.0}$       | $53.9_{↑0.9}$      | $37.0_{↓2.0}$       |
> > | GPT-4o                    | $61.4_{↓0.3}$      | $44.9_{↓1.5}$       | $51.5_{↓1.7}$      | $37.7_{↓4.9}$      |
> > | DeepSeek-V3               | $51.9_{↓6.0}$      | $42.2_{↑2.9}$       | $54.4_{↑0.6}$      | $37.0_{↓4.1}$       |
> > | DeepSeek-R1               | $54.8_{↓4.8}$      | $41.4_{↓1.5}$       | $60.0_{↑2.2}$      | $44.1_{↑1.7}$      |
> >
> > - For the scalar-based and generative LLM-based comparison, we argue that it is the underlying difference between traditional and modern metrics; we cannot let traditional metrics generate which one is better in natural languages. To ensure a fair and controlled comparison, we introduce the following experimental condition: we prompt the LLMs to act as scalar-based evaluators. Specifically, we instruct them to output only a final score for each response, which directly mirrors the process of traditional evaluation metrics. The results are listed below, where we find that the performance degrades for all the models across settings, indicating that pairwise evaluation is easier than pointwise evaluation for LLMs, further validating our assumption that LLMs cannot evaluate long-form responses well.
> >
> > | Model                     | LFQA-E-EN       |                 | LFQA-E-ZH       |                 |
> > |---------------------------|-----------------|-----------------|-----------------|-----------------|
> > |                           | Acc             | F1              | Acc             | F1              |
> > | Qwen2.5-32B-Instruct      | $45.2_{↓18.3}$      | $38.4_{↓7.4}$       | $40.0_{↓16.7}$      | $36.0_{↓5.8}$       |
> > | Qwen2.5-72B-Instruct      | $37.6_{↓23.6}$      | $33.2_{↓9.9}$       | $38.9_{↓14.1}$      | $36.0_{↓3.0}$       |
> > | GPT-4o                    | $49.3_{↓12.4}$      | $38.0_{↓8.4}$       | $43.7_{↓9.5}$       | $37.8_{↓4.8}$       |
> > | DeepSeek-V3               | $36.9_{↓21.0}$      | $33.6_{↓5.7}$       | $43.7_{↓10.1}$      | $37.4_{↓3.7}$       |
> > | DeepSeek-R1               | $49.9_{↓9.7}$       | $40.5_{↓2.4}$       | $50.5_{↓7.3}$       | $42.9_{↑0.5}$       |

---

> ### Author Response · Authors · 2025-11-19
>
> **(O4)**  Thank you for this valuable suggestion. Our primary goal was to benchmark the core evaluation capabilities of models across different architectures (e.g., reward models, LLMs, LRMs) without additional techniques. However, we agree that including more recent, specialized methods strengthens our work's comprehensiveness. To address this, we have now incorporated several additional baselines, including factuality-oriented and evaluation-ensemble measures, as shown below. Through the experimental results, we find that the agentic framework is the most effective method for LFQA evaluation, and majority voting also contributes to improvement.
>
> | Model                     | LFQA-E-EN       |                 | LFQA-E-ZH       |                 |
> |---------------------------|-----------------|-----------------|-----------------|-----------------|
> |                           | Acc             | F1              | Acc             | F1              |
> | SOTA Models               |                 |                 |                 |                 |
> | Gamini2.5-flash           | 63.3            | 45.6            | 57.9            | 44.7            |
> | GPT-5                     | 64.8            | 47.7            | 60.1            | 46.7            |
> | FactScore variants        |                 |                 |                 |                 |
> | FineSurE (GPT-4o)         | 47.8            | 35.5            | 42.9            | 32.5            |
> | Evaluation Ensembles      |                 |                 |                 |                 |
> | ChatEval (GPT-4o)         | 70.7            | 48.0            | 49.9            | 40.3            |
> | G-Eval (GPT-4o)           | 68.0            | 50.0            | 55.2            | 42.2            |
>
> **Question**
>
> Please see O1 - O4
>
> **References**
>
> [1] Fangyuan Xu, Yixiao Song, Mohit Iyyer, and Eunsol Choi. 2023. A Critical Evaluation of Evaluations for Long-form Question Answering. arXiv preprint arXiv:2305.18201.

---

### Author Response · Authors · 2025-12-03
**Discussion Overview for Area Chair**

Dear Area Chair,

We sincerely appreciate your efforts during this challenging period. We understand the exceptional workload you're managing, and we're grateful for your careful consideration of our work. To further assist you in gaining a deeper understanding of the manuscript, we would like to provide a summary of our recent discussions with the reviewers.

## Reviewer Assessment Overview

All reviewers recognized the timeliness and methodology of our work, particularly in introducing the LFQA-E benchmark for evaluating LFQA systems. While three reviewers (awMT, Yu2f, and H1wd) gave scores of 8, 4, and 4, reviewer aZgf gave a rating of 2, citing concerns about the comparison to prior work in the field.

After intensive and concise discussion, we believe that the feedback provided by the reviewers has been extremely valuable, and we have addressed all their concerns comprehensively in our response.

---

## Reviewer awMT

### 1. Top-ranked answers from Reddit introduce quality variance
We clarified that experts do not edit Reddit content; instead, they strictly filter out low-quality or irrelevant QA pairs during dataset curation. Specifically, experts reviewed 2,236 ELI5 QA pairs and retained only 1,026 high-quality pairs. We further provided quantitative expert quality ratings (completeness, factuality, professionalism, useful examples, clarity), showing substantial improvements after filtering (e.g., factuality improved from 2.8 to 4.6). High Cohen’s Kappa also confirms strong inter-expert consistency.

### 2. Annotation Cost, Efficiency, and Annotator Qualifications
The reviewer questioned whether 2 dollars per question risked rushed annotations. We clarified that this rate exceeds the local minimum wage and corresponds to 3–6 per hour. Each comparison takes ~7 minutes. Additionally, we provided detailed annotator qualification statistics, showing that most annotators hold PhD or MS degrees across diverse disciplines, with high historical consistency (80–90%), and undergo a preliminary training phase.

### 3. Experimental Controls and Prompt Sensitivity
We clarified that all LLM- and LRM-based evaluators use the same prompt template, except for fine-tuned metrics by necessity. We further conducted new prompt-sensitivity ablations (with and without CoT) and scalar-vs-pairwise evaluation experiments, showing consistent performance degradation without CoT and under scalar scoring.

### 4. Missing Recent Evaluation Baselines
We added FactScore variants, ChatEval, and G-Eval as new baselines. The additional results show that agentic and ensemble-based evaluation performs best, reinforcing rather than weakening our original conclusions.

---

## Reviewer Yu2f

### 1. Missing Comparison with Prior Benchmarks
We conducted additional large-scale comparative experiments against established LFQA and reward-model benchmarks, including the expert-annotated benchmark from Xu et al., Feedback-Bench, RM-Bench, and Reward-Bench, using multiple frontier models (GPT-5, Qwen2.5, DeepSeek). Results consistently show a substantial performance drop on LFQA-E, demonstrating that LFQA-E is significantly more challenging and better differentiates state-of-the-art models.

### 2. Difficulty, Multilingual Plan, and Licensing
We clarified that LFQA-E difficulty is validated empirically via performance gaps among SOTA models rather than a single scalar metric. We confirmed plans to release a multilingual version and a public leaderboard, and we detailed our compliance with Reddit licensing restrictions for public release.

---

## Reviewer H1wd

### 1. Length and Definition of "Long-Form"
We clarified that Table 1 reports word counts, not tokens, and that LFQA-E aligns with human expert behavior in ELI5, where the average high-quality answer is ~215 words. We also demonstrated that the true difficulty lies in evaluation complexity rather than output length: each evaluation instance requires processing *Question + Reference + Two Candidates*, resulting in ~1,300-token inputs. This makes LFQA-E substantially more demanding than generation-only benchmarks and emphasizes reasoning density over raw context length.

### 2. Use of Intermediate Reasoning (CoT)
We confirmed that our LLM- and LRM-based evaluators explicitly use Chain-of-Thought prompting, with concrete evidence in multiple case studies, directly addressing the reviewer’s concern.

### 3. Thinking Budget vs. Accuracy
We conducted additional Pass@K experiments with a strong reasoning model, showing a clear, quantifiable relationship between sampling budget and evaluation accuracy.

### 4. Distinction from Reward-Model Benchmarks
We demonstrated that LFQA-E differs fundamentally through expert-written references, multilingual design, and significantly higher difficulty, supported by consistent performance drops relative to RewardBench and RM-Bench.

---

> ### Author Response · Authors · 2025-12-03
>
> ## Reviewer aZgf
>
> ### 1. Reference-Based Evaluation Validity
> We justified the necessity of references as an objective grounding that prevents metrics from rewarding fluent but incorrect answers. We emphasized that all references are expert-written to be comprehensive, and that modern LLMs can retrieve information before evaluation in real-world deployments, making reference-aware evaluation realistic and controlled.
>
> ### 2. Annotation Reliability and Preference Semantics
> We clarified that labels come from a pool of 10 professional annotators, not just two. When two annotators disagree, a third adjudicator determines the final label. The reported κ = 0.77 reflects overall pool consistency. We further conducted a post-hoc factor analysis over 600 comparisons, showing preferences are driven mainly by factuality and completeness, not stylistic bias.
>
> ### 3. Distinction from Prior Benchmarks
> We clarified that LFQA-E differs fundamentally via expert-driven manual curation (not auto-collected), intentionally difficult comparisons, and a multilingual, multi-setting design. We conducted additional experiments to show that frontier models achieve very high scores on Feedback-Bench but drop sharply on LFQA-E (e.g., GPT-4o: 89.2% → 57.5%), demonstrating substantially stronger discriminative power.
>
> ---
>
> We hope that the reviewers' concerns have now been fully addressed and that the manuscript can be considered for acceptance. We are grateful for the thoughtful feedback provided by the reviewers and for your continued support during this process.
>
> Sincerely,
> **ICLR 2026 Conference Submission 6690 Authors**

---

### Meta-Review · Area_Chair_a3h3 · 2026-01-06

**Summary:**

This paper introduces LFQA-E, a reference-based multilingual benchmark for evaluating automatic metrics on long-form question answering (LFQA). The motivation is that existing automatic metrics fall short on LFQA, and also there are gaps in metric generalization. The benchmark contains questions and pairwise comparisons across 15 topics in English and Chinese, with expert-written references and a triple-choice annotation protocol (A / B / tie). The authors evaluate 17 evaluation methods spanning static, LLM-based, reward/model-based, LRM, and trained evaluation models.

Reviewers raised concerns about the quality of the reference answers (from Reddit), the potential for annotator pay rates to lead to rushed judgments, inconsistent experimental controls, missing recent baseline methods, comparisons with existing LFQA benchmarks using frontier models, clarification of difficulty metrics, and details about multilingual release plans and licensing compliance. Reviewer H1wd  challenged whether 180-200 words constitutes "long-form" given modern LLM capabilities, asked about chain-of-thought prompting in evaluators, requested analysis of thinking budget versus accuracy, and sought clarification on distinctions from existing reward model benchmarks. Most critically, Reviewer aZgf
  argued the work lacked substantial novelty compared to Feedback-Bench, criticized reference-based evaluation as unrealistic for real-world queries, raised concerns about annotation noise with only two annotators per question achieving κ=0.77, and questioned whether preferences captured meaningful aspects beyond annotator bias.

In my opinion, the benchmark will be of value to the research community, not only for the development of QA evaluation metrics but because (as they authors state in their response to a reviewer comment) it allows us to work on rewards for non-verifiable tasks.
The authors addressed the reviewers' concerns through additional experiments, clarifications about expert curation processes, comprehensive comparisons showing LFQA-E is significantly more challenging than existing benchmarks, and detailed annotator qualification statistics (see below for details).

**Reviewer Concerns:**

1.  Reference quality from Reddit, low annotator pay, inconsistent experimental controls, and missing baselines. Authors responded by clarifying that experts filter (not edit) Reddit content, retaining only 1,026 of 2,236 pairs, and that quality ratings showed substantial improvements. They demonstrated that $2/question exceeds the local minimum wage, provided detailed annotator qualifications (mostly PhD/MS holders with 80-90% consistency), confirmed that all LLM evaluators use identical prompts, and added FactScore variants, ChatEval, and G-Eval as new baselines, showing that agentic evaluation performs best.

2. A reviewer requested comparisons with prior benchmarks and clarification on difficulty metrics, multilingual plans, and licensing. Authors responded with extensive comparison experiments showing frontier models achieve high scores on Feedback-Bench and Reward-Bench but drop significantly on LFQA-E, demonstrating substantially greater difficulty. They confirmed plans for multilingual release, public leaderboard, and Reddit license compliance.

3. There were questions about whether a 180-200-word answer qualifies as "long-form," about chain-of-thought usage, and what distinguishes the proposed benchmark from reward benchmarks. Authors clarified that Table 1 reports words (not tokens), aligning with ELI5's 215-word average, and that evaluation complexity comes from processing ~1,300 tokens (Question + Reference + Two Candidates). They confirmed CoT prompting is used (evidenced in case studies), conducted Pass@K experiments showing clear accuracy-budget relationships, and demonstrated LFQA-E's unique features: expert-written references, multilingual design, and significantly greater difficulty than existing reward benchmarks.

4.  Limited novelty versus Feedback-Bench. The authors responded by demonstrating LFQA-E's manual curation (vs. Feedback-Bench's automated collection), showing performance drops of 30-40 percentage points compared to Feedback-Bench, justifying references as objective grounding preventing reward of fluent but incorrect answers.

**Reviewer Scores:**

The authors made substantial additions, including new baseline comparisons, resolution ablations, details on annotator qualification, and comprehensive benchmark comparisons that demonstrate LFQA-E's significantly greater challenge and discriminative power. I am not sure the 2 would have gone up, but the 4s could be 6s, as all the issues raised were answered.

---

### Decision · Program_Chairs · 2026-01-26

Accept (Poster)